# End-to-End Autonomous Driving without Costly Modularization and 3D Manual Annotation

## Abstract

We propose UAD, a method for vision-based end-to-end autonomous driving (E2EAD), achieving the best open-loop evaluation performance in nuScenes, meanwhile showing robust closed-loop driving quality in CARLA. Our motivation stems from the observation that current E2EAD models still mimic the modular architecture in typical driving stacks, with carefully designed **supervised** perception and prediction subtasks to provide environment information for oriented planning. Although achieving groundbreaking progress, such design has certain drawbacks: 1) preceding subtasks require massive high-quality 3D annotations as supervision, posing a significant impediment to scaling the training data; 2) each submodule entails substantial computation overhead in both training and inference. To this end, we propose UAD, an E2EAD framework with an **unsupervised**[1] proxy to address all these issues. Firstly, we design a novel Angular Perception Pretext to eliminate the annotation requirement. The pretext models the driving scene by predicting the angular-wise spatial objectness and temporal dynamics, without manual annotation. Secondly, a self-supervised training strategy, which learns the consistency of the predicted trajectories under different augment views, is proposed to enhance the planning robustness in steering scenarios. Our UAD achieves 38.7% relative improvements over UniAD on the average collision rate in nuScenes and surpasses VAD for 6.40 points on the driving score in CARLA's Town05 Long benchmark. Moreover, the proposed method only consumes 44.3% training resources of UniAD and runs $3.4\times$ faster in inference. Our innovative design not only for the first time demonstrates unarguable performance advantages over supervised counterparts, but also enjoys unprecedented efficiency in data, training, and inference.

## 1  Introduction

Recent decades have witnessed breakthrough achievements in autonomous driving. The end-to-end paradigm, which seeks to integrate perception, prediction, and planning tasks into a unified framework, stands as a representative branch [33, 1, 39, 3, 35, 21, 22]. The latest advances in end-to-end autonomous driving significantly piqued researchers' interest [21, 22]. However, handcrafted and resource-intensive supervised sub-tasks for perception and prediction, which have previously proved their utility in environment modeling [35, 3, 20], continue to be indispensable, as shown in Fig. 1a.

Then what insights have we gained from the recent advances? It has come to our attention that one of the most enlightening innovations lies in the Transformer-based pipeline, in which the queries act as a connective thread, seamlessly bridging various tasks. Besides, the capability for environment modeling has also seen a significant boost, primarily due to complicated interactions of supervised

---

[1]Following [30, 4], here we consider the methods as "unsupervised" ones as long as no manual annotation is used and required in the target task or domain.

Submitted to 38th Conference on Neural Information Processing Systems (NeurIPS 2024). Do not distribute.

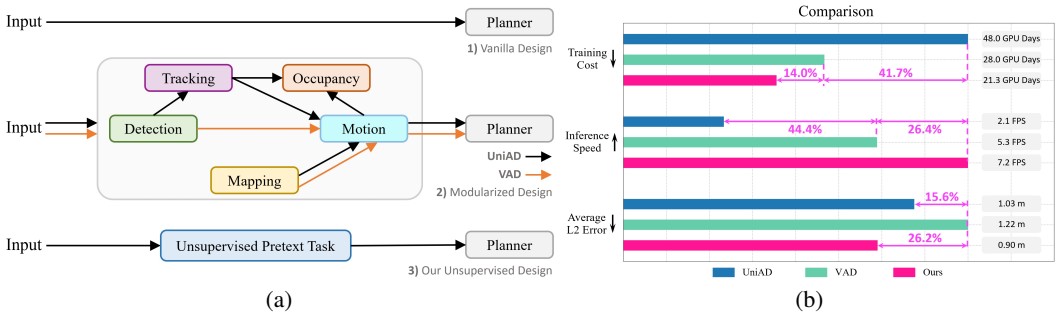

Figure 1: **(a)** End-to-end autonomous driving paradigms. 1) The vanilla architecture that directly predicts control command. 2) The modularized design that combines various preceding tasks. 3) Our proposed framework with unsupervised pretext task. **(b)** Comparison of training cost, inference speed and average L2 error between our method and [21, 22] on 8 NVIDIA Tesla A100 GPUs.

sub-tasks. However, every coin has two sides. In comparison to the vanilla design [33] (see Fig. 1a), modularized methods incur unavoidable computation and annotation overhead. As illustrated in Fig. 1b, the training of the recent method UniAD [21] takes 48 GPU days while running at only 2.1 frames per second (FPS). Moreover, modules in existing perception and prediction design require large quantities of high-quality annotated data. The financial overhead for human annotation significantly impedes the scalability of such modularized methods with supervised subtasks to leverage massive data. As proved by large foundation models [24, 31], scaling up the data volume is the key to bringing the model capabilities to the next level. Thus we ask ourselves the question: *Is it viable to devise an efficient and robust E2EAD framework while alleviating the reliance on 3D annotation?*

In this work, we show the answer is affirmative by proposing an innovative **U**nsupervised pretext task for end-to-end **A**utonomous **D**riving (**UAD**), which seeks to efficiently model the environment. The pretext task consists of an angular-wise perception module to learn *spatial* information by predicting the objectness of each sector region in BEV space, and an angular-wise dreaming decoder to absorb *temporal* knowledge by predicting inaccessible future states. The introduced angular queries link the two modules as a whole pretext task to perceive the driving scene. Notably, our method shines by completely eliminating the annotation requirement for perception and prediction. Such data efficiency is not attainable for current methods with complex supervised modularization [21, 22]. The supervision for learning spatial objectness is obtained by projecting the 2D region of interests (ROIs) from an off-the-shelf open-set detector [28] to BEV space. While utilizing the publicly available open-set 2D detector pre-trained with manual annotation from other domains (*e.g.* COCO [27]), we avoid the need for any additional 3D labels within our paradigm and target domains (*e.g.* nuScenes [2] and CARLA [11]), thereby creating a pragmatically unsupervised setting [30]. Furthermore, we introduce a self-supervised direction-aware learning strategy to train the planning model. Specifically, the visual observations are augmented with different rotation angles, and the consistency loss is applied to the predictions for robust planning. Without bells and whistles, the proposed UAD outperforms UniAD for 0.13m in nuScenes Avg. L2 error, and surpasses VAD [22] for 9.92 points in CARLA route completion score. Such unprecedented performance gain is achieved with a $3.4\times$ inference speed, a mere 44.3% training budget of UniAD, and zero annotations, as illustrated in Fig. 1b.

In summary, our contributions are as follows: **1)** We propose an unsupervised pretext task to discard the requirement of 3D manual annotation in end-to-end autonomous driving, potentially making it more feasible to scale the training data to billions level without any labeling overload; **2)** We introduce a novel self-supervised direction-aware learning strategy to maximize the consistency of the predicted trajectories under different augment views, which enhances planning robustness in steering scenarios; **3)** Our method shows superiority in both open- and closed-loop evaluation compared with other vision-based E2EAD methods, with much lower computation and annotation cost.

## 2 Related Work

### 2.1 End-to-End Autonomous Driving

End-to-end autonomous driving can be dated back to 1988, when the ALVINN [33] proposed by Carnegie Mellon University could successfully navigate a vehicle over 400 meters. After that, to

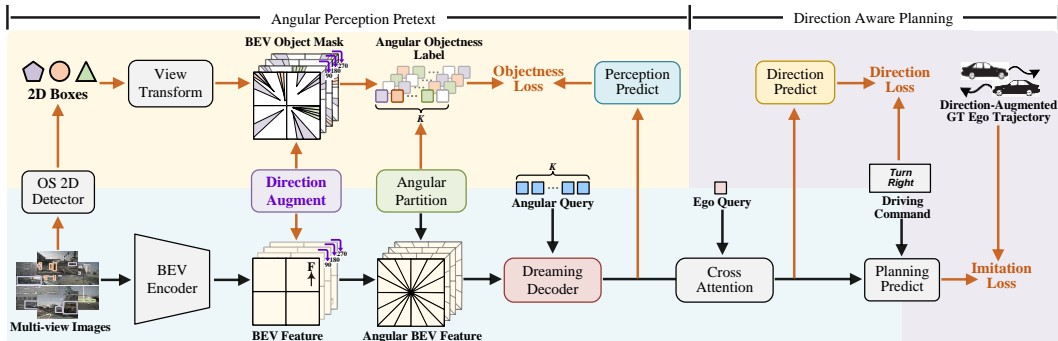

Figure 2: The architecture of our UAD. The inference pipeline is marked by black arrows with blue background, which plans ego trajectory based on the input multi-view images. The training pipeline consists of Angular Perception Pretext (orange arrows with khaki background) and Direction-Aware Planning (orange arrows with purple background). "F" in BEV feature indicates the driving direction.

improve the robustness of E2EAD, a series of modern approaches such as NEAT [6], P3 [35], MP3 [3], ST-P3 [20] introduce the design of more dedicated modularization, which integrate auxiliary information such as HD maps, and additional tasks like bird's-eye view (BEV) segmentation. Most recently, embracing advanced architectures like Transfromer [37] and visual occupancy prediction [29], UniAD [21] and VAD [22] demonstrate impressive performance in open-loop evaluation. In this work, instead of integrating complex supervised modular sub-tasks, we innovatively propose another path proving that an efficient unsupervised pretext task without any human annotation like 3D bounding boxes and point cloud categories, can achieve even superior performance than recent state-of-the-arts.

## 2.2 World Model

In pursuit of understanding the dynamic changes in environments, researchers in the fields of gaming and robotics have proposed various world models [13, 14, 15, 16]. Recently, the autonomous driving community introduces world models for safer maneuvering [32, 18, 12, 38]. MILE [18] considers the environment as a high-level embedding and tends to predict its future state with historical observations. Drive-WM [38] proposes a framework to integrate world models with existing E2E methods to improve planning robustness. In this work, we propose an auto-regressive mechanism, tailored to our unsupervised pretext, to capture angular-wise temporal dynamics within each sector.

## 3 Method

### 3.1 Overview

As illustrated in Fig. 2, our **UAD** framework consists of two essential components: 1) the Angular Perception Pretext, aims to liberate E2EAD from costly modularized tasks in an unsupervised fashion; 2) the Direction-Aware Planning, learns self-supervised consistency of the augmented trajectories.

Specifically, UAD first models the driving environment with the pretext. The *spatial* knowledge is acquired by estimating the objectness of each sector region within the BEV space. The angular queries, each responsible for a sector, are introduced to extract features and predict the objectness. The supervision label is generated by projecting the 2D regions of interests (ROIs) to the BEV space, which are predicted with an available open-set detector GroundingDINO [28]. This way not only eliminates the 3D annotation requirement, but also greatly reduces the training budget. Moreover, as driving is inherently a dynamic and continuous process, we thus propose an angular-wise dreaming decoder to encode the *temporal* knowledge. The dreaming decoder can be viewed as an augmented world model [13] capable of auto-regressively predicting the future states.

Subsequently, direction-aware planning is introduced to train the planning module. The raw BEV feature is augmented with different rotation angles, yielding rotated BEV representations and ego trajectories. We apply self-supervised consistency loss to the predicted trajectories of each augmented view, which is expected to improve the robustness for directional change and input noises. The learning strategy can also be regarded as a novel data augmentation technique customized for end-to-end autonomous driving, which enhances the diversity of trajectory distribution.

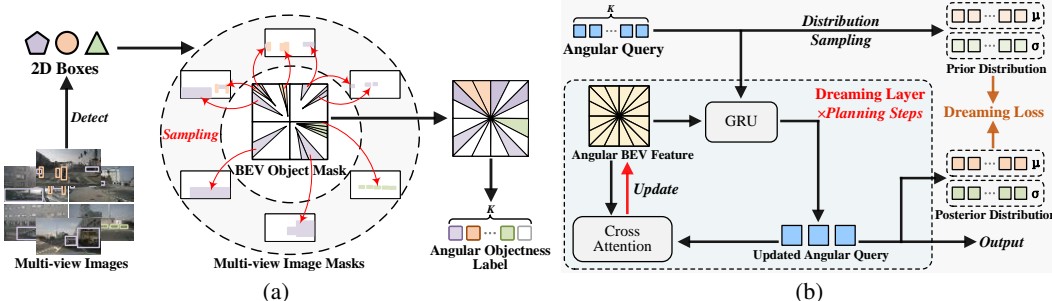

Figure 3: **(a)** Label generation for angular perception pretext. **(b)** Illustration of dreaming decoder.

## 3.2 Angular Perception Pretext

**Spatial Representation Learning.** Our model attempts to acquire spatial knowledge of the driving scene by predicting the objectness of each sector region within the BEV space. Specifically, taking multi-view images $\{\mathbf{I}_i \in \mathbb{R}^{H_i \times W_i \times 3}\}$ as input, the BEV encoder [25] first extracts visual information into the BEV feature $\mathbf{F}_b \in \mathbb{R}^{H_b \times W_b \times C}$. Then, $\mathbf{F}_b$ is partitioned into $K$ sectors with a uniform angle $\theta$ centered around ego car. Each sector contains several feature points in BEV space. Denoting feature of a sector as $\mathbf{f} \in \mathbb{R}^{N \times C}$, where $N$ is the maximum number of feature points in all sectors, we derive angular BEV feature $\mathbf{F}_a \in \mathbb{R}^{K \times N \times C}$. Zero-padding is applied on sectors with fewer than $N$ points.

Then why do we partition the rectangular BEV feature to angular-wise formatting? The underlying reason is that, in the absence of depth information, the region in BEV space corresponding to an ROI in 2D image is a sector. As illustrated in Fig. 3a, by projecting 3D sampling points to images and verifying their presence in 2D ROIs, a BEV object mask $\mathbf{M} \in \mathbb{R}^{H_b \times W_b \times 1}$ is generated, representing the objectness in BEV space. Specifically, the sampling points falling within 2D ROIs are set to 1, while the others are 0. It is noticed that the positive sectors are irregularly and sparsely distributed in BEV space. To make the objectness label more compact, similar to the BEV feature partition, we uniformly divide $\mathbf{M}$ into $K$ equal parts. The segments overlapped with positive sectors are assigned with 1, constituting the angular objectness label $\mathbf{Y}_{obj} \in \mathbb{R}^{K \times 1}$. Thanks to the rapid development of open-set detection, it's now convenient to obtain 2D ROIs for the input multi-view images by feeding the pre-defined prompts (*e.g.,* vehicle, pedestrian, and barrier) to a 2D open-set detector like GroundingDINO [28]. Such design is the key in reducing annotation cost and scaling up the dataset.

To predict the objectness score of each sector, we define angular queries $\mathbf{Q}_a \in \mathbb{R}^{K \times C}$ to summarize $\mathbf{F}_a$. Each angular query $\mathbf{q}_a \in \mathbb{R}^{1 \times C}$ in $\mathbf{Q}_a$ will interact with corresponding $\mathbf{f}$ by cross attention [37],

$$\mathbf{q}_a = \mathrm{CrossAttention}(\mathbf{q}_a, \mathbf{f}), \tag{1}$$

Finally, we map $\mathbf{Q}_a$ to the objectness scores $\mathbf{P}_a \in \mathbb{R}^{K \times 1}$ with a linear layer, which is supervised by $\mathbf{Y}_{obj}$ with binary cross-entropy loss (denoted as $\mathcal{L}_{spat}$).

**Temporal Representation Learning.** We propose to capture the temporal information of driving scenarios with the angular-wise dreaming decoder. As shown in Fig. 3b, the decoder auto-regressively learns transition dynamics of each sector in a similar way of world model [14]. Assuming the planning module predicts the trajectories of future $T$ steps, the dreaming decoder accordingly comprises $T$ layers, where each updates the input angular queries $\mathbf{Q}_a$ and angular BEV feature $\mathbf{F}_a$ based on the learned temporal dynamics. At step $t$, the queries $\mathbf{Q}_a^{t-1}$ first grasp environmental dynamics from the observation feature $\mathbf{F}_a^t$ with a gated recurrent unit (GRU) [7], which generates $\mathbf{Q}_a^t$ (hidden state),

$$\mathbf{Q}_a^t = \mathrm{GRU}(\mathbf{Q}_a^{t-1}, \mathbf{F}_a^t), \tag{2}$$

In previous world models, the hidden state $\mathbf{Q}$ is solely used for perceiving observed scenes. The GRU iteration thus ends at $t$ with the final observation $\mathbf{F}_a^t$. In our framework, $\mathbf{Q}$ is also used for predicting ego trajectories in the future. Yet, the future observation, *e.g.*, $\mathbf{F}_a^{t+1}$, is unavailable, as the world model [14] is designed for forecasting the future with only current observation. To obtain $\mathbf{Q}_a^{t+1}$, we first propose to update $\mathbf{F}_a^t$ to provide pseudo observations $\hat{\mathbf{F}}_a^{t+1}$,

$$\hat{\mathbf{F}}_a^{t+1} = \mathrm{CrossAttention}(\mathbf{F}_a^t, \mathbf{Q}_a^t). \tag{3}$$

Then $\mathbf{Q}_a^{t+1}$ can be generated with Eq. 2 and inputs of $\hat{\mathbf{F}}_a^{t+1}$ and $\mathbf{Q}_a^t$.

Following the loss design in world models [14, 15, 16], we respectively map $\mathbf{Q}_a^{t-1}$ and $\mathbf{Q}_a^t$ to distributions of $\{\mu_a^{t-1}, \sigma_a^{t-1} \in \mathbb{R}^{K \times C}\}$ and $\{\mu_a^t, \sigma_a^t \in \mathbb{R}^{K \times C}\}$, and then minimize their KL divergence.

For the prior distribution from $\mathbf{Q}_a^{t-1}$, it's regarded as a prediction of the future dynamics without observation. In contrast, the posterior distribution from $\hat{\mathbf{Q}}_a^t$ represents the future dynamics with the observation $\mathbf{F}_a^t$. The KL divergence between the two distributions measures the gap between the imagined future (prior) and the true future (posterior). We expect to enhance the capability of future prediction for long-term driving safety, which is realized by optimizing the dreaming loss $\mathcal{L}_{\mathrm{drm}}$,

$$\mathcal{L}_{\mathrm{drm}} = \mathrm{KL}(\{\mu_a^t, \sigma_a^t\}||\{\mu_a^{t-1}, \sigma_a^{t-1}\}), \tag{4}$$

### 3.3 Direction-Aware Planning

**Planning Head.** The outputs of angular perception pretext contain a group of angular queries $\{\mathbf{Q}_a^t \ (t = 1, ..., T)\}$. For planning, we correspondingly initialize $T$ ego queries $\{\mathbf{Q}_{\mathrm{ego}}^t \in \mathbb{R}^{1 \times C} \ (t = 1, ..., T)\}$ to extract planning-relevant information and predict the ego trajectory of each future time step. The interaction between ego queries and angular queries is performed with cross attention,

$$\mathbf{Q}_{\mathrm{ego}}^t = \mathrm{CrossAttention}(\mathbf{Q}_{\mathrm{ego}}^t, \mathbf{Q}_a^t). \tag{5}$$

The output ego queries $\{\mathbf{Q}_{\mathrm{ego}}^t\}$ are then used to predict the ego trajectories of future $T$ steps. Following previous works [21, 22], a high-level driving signal $c$ (*turn left*, *turn right* or *go straight*) is provided as prior knowledge. The planning head takes the concatenated ego feature $\mathbf{F}_{\mathrm{ego}} \in \mathbb{R}^{T \times C}$ from $\{\mathbf{Q}_{\mathrm{ego}}^t\}$ and the driving command $c$ as inputs, and outputs the planning trajectory $\mathbf{P}_{\mathrm{traj}} \in \mathbb{R}^{T \times 2}$,

$$\mathbf{P}_{\mathrm{traj}} = \mathrm{PlanHead}(\mathbf{F}_{\mathrm{ego}}, c), \tag{6}$$

where the $\mathrm{PlanHead}$ is the same as UniAD [21]. We apply $\mathcal{L}_1$ loss to minimize the distance between the predicted ego trajectory $\mathbf{P}_{\mathrm{traj}}$ and the ground truth $\mathbf{G}_{\mathrm{traj}}$, denoted as $\mathcal{L}_{\mathrm{imi}}$. Notably, $\mathbf{G}_{\mathrm{traj}}$ is easy to obtain, and manual annotation is not required in practical scenarios.

**Directional Augmentation.** Observed that the training data is predominated by the *go straight* scenarios, we propose a directional augmentation strategy to balance the distribution. As shown in Fig. 4, the BEV feature $\mathbf{F}_b$ is rotated with different angles $r \in R = \{90°, 180°, 270°\}$, yielding the rotated representations $\{\mathbf{F}_b^r\}$. The augmented features will also be used for the pretext and planning task, and supervised by the aforementioned loss functions (*e.g.*, $\mathcal{L}_{\mathrm{spat}}$). Notably, the BEV object mask $\mathbf{M}$ and the ground truth ego trajectory $\mathbf{G}_{\mathrm{traj}}$ are also rotated to provide corresponding supervision labels.

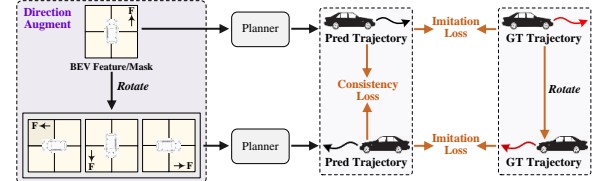

Figure 4: Illustration of direction-aware learning strategy.

Furthermore, we propose an auxiliary task to enhance the steering capability. In specific, we predict the planning direction that the ego car intends to maneuver (*i.e., left*, *straight* or *right*) based on the ego query $\mathbf{Q}_{\mathrm{ego}}^t$, which is mapped to the probabilities of three directions $\mathbf{P}_{\mathrm{dir}}^t \in \mathbb{R}^{1 \times 3}$. The direction label $\mathbf{Y}_{\mathrm{dir}}^t$ is generated by comparing the x-axis value of ground truth $\mathbf{G}_{\mathrm{traj}}^t(x)$ with the threshold $\delta$. Specifically, $\mathbf{Y}_{\mathrm{dir}}^t$ is assigned to $straight$ if $-\delta < \mathbf{G}_{\mathrm{traj}}^t(x) < \delta$, otherwise $\mathbf{Y}_{\mathrm{dir}}^t = left/right$ for $\mathbf{G}_{\mathrm{traj}}^t(x) \leqslant -\delta / \mathbf{G}_{\mathrm{traj}}^t(x) \geqslant \delta$, respectively. We use the cross-entropy loss to minimize the gap between the direction prediction $\mathbf{P}_{\mathrm{dir}}^t$ and the direction label $\mathbf{Y}_{\mathrm{dir}}^t$, denoted as $\mathcal{L}_{\mathrm{dir}}$.

**Directional Consistency.** Tailored to the introduced directional augmentation, we propose a directional consistency loss to improve the augmented plan training in a self-supervised manner. It should be noticed that the augmented trajectory predictions $\mathbf{P}_{\mathrm{traj}}^{t,r}$ incorporate the same scene information as the original one $\mathbf{P}_{\mathrm{traj}}^{t,r=0}$, *i.e.,* BEV features with different rotation angles. Therefore, it's reasonable to consider the consistency among the predictions and regulate the noises caused by the rotation. The planning head is expected to be more robust to directional change and input distractors. Specifically, $\mathbf{P}_{\mathrm{traj}}^{t,r}$ are first rotated back to the original scene direction, then $\mathcal{L}_1$ loss is applied with $\mathbf{P}_{\mathrm{traj}}^{t,r=0}$,

$$\mathcal{L}_{\mathrm{cons}} = \frac{1}{T \cdot |R|} \sum_{t=1}^{T} \sum_r^R ||\mathrm{Rot}(\mathbf{P}_{\mathrm{traj}}^{t,r}) - \mathbf{P}_{\mathrm{traj}}^{t,r=0}||_1, \tag{7}$$

where $\mathrm{Rot}$ is the inverse rotation.

To summarize, the overall objective for our UAD contains spatial objectness loss, dreaming loss from the pretext, and imitation learning loss, direction loss, consistency loss from the planning task,

$$\mathcal{L} = \omega_1 \mathcal{L}_{\mathrm{spat}} + \omega_2 \mathcal{L}_{\mathrm{drm}} + \omega_3 \mathcal{L}_{\mathrm{imi}} + \omega_4 \mathcal{L}_{\mathrm{dir}} + \omega_5 \mathcal{L}_{\mathrm{cons}}, \tag{8}$$

where $\omega_1, \omega_2, \omega_3, \omega_4, \omega_5$ are the weight coefficients.

Table 1: Open-loop planning performance in nuScenes [2]. [†] indicates LiDAR-based method and [‡] denotes TemAvg evaluation protocol used in VAD and ST-P3 (see Eq. 9 for details). [◊] means using ego status in the planning module and calculating collision rates following BEV-Planner [26].

| Method | Tasks with 3D annotation | L2 (m) ↓ | | | | Collision (%) ↓ | | | | Intersection (%) ↓ | | | | FPS |
|---|---|---|---|---|---|---|---|---|---|---|---|---|---|---|
| | | 1s | 2s | 3s | Avg. | 1s | 2s | 3s | Avg. | 1s | 2s | 3s | Avg. | |
| NMP[†] [39] | Det & Motion | - | - | 2.31 | - | - | - | 1.92 | - | - | - | - | - | - |
| SA-NMP[†] [39] | Det & Motion | - | - | 2.05 | - | - | - | 1.59 | - | - | - | - | - | - |
| FF[†] [19] | Occ | 0.55 | 1.20 | 2.54 | 1.43 | 0.06 | 0.17 | 1.07 | 0.43 | - | - | - | - | - |
| EO[†] [23] | Occ | 0.67 | 1.36 | 2.78 | 1.60 | 0.04 | 0.09 | 0.88 | 0.33 | - | - | - | - | - |
| ST-P3 [20] | Det & Map | 1.72 | 3.26 | 4.86 | 3.28 | 0.44 | 1.08 | 3.01 | 1.51 | 2.53 | 8.17 | 14.4 | 8.37 | 1.8 |
| UniAD [21] | Det&Track&Map&Motion&Occ | 0.48 | 0.96 | 1.65 | 1.03 | 0.05 | 0.17 | 0.71 | 0.31 | 0.21 | 1.32 | 3.63 | 1.72 | 2.1 |
| VAD-Tiny [22] | Det & Map & Motion | 0.60 | 1.23 | 2.06 | 1.30 | 0.33 | 1.33 | 2.21 | 1.29 | 0.94 | 3.22 | 7.65 | 3.94 | 17.6 |
| VAD-Base [22] | Det & Map & Motion | 0.54 | 1.15 | 1.98 | 1.22 | 0.10 | 0.24 | 0.96 | 0.43 | 0.60 | 2.38 | 5.18 | 2.72 | 5.3 |
| OccNet [36] | Det & Map & Occ | 1.29 | 2.13 | 2.99 | 2.14 | 0.21 | 0.59 | 1.37 | 0.72 | - | - | - | - | 3.3 |
| UAD-Tiny (Ours) | None | 0.47 | 0.99 | 1.71 | 1.06 | 0.08 | 0.39 | 0.90 | 0.46 | 0.24 | 1.15 | 3.12 | 1.50 | 18.9 |
| UAD (Ours) | None | 0.39 | 0.81 | 1.50 | 0.90 | 0.01 | 0.12 | 0.43 | 0.19 | 0.13 | 0.88 | 2.66 | 1.22 | 7.2 |
| ST-P3[‡] [20] | Det & Map | 1.33 | 2.11 | 2.90 | 2.11 | 0.23 | 0.62 | 1.27 | 0.71 | 2.53 | 8.17 | 14.4 | 8.37 | 1.8 |
| UniAD[‡] [21] | Det&Track&Map&Motion&Occ | 0.44 | 0.67 | 0.96 | 0.69 | 0.04 | 0.08 | 0.23 | 0.12 | 0.21 | 1.32 | 3.63 | 1.72 | 2.1 |
| VAD-Base[‡] [22] | Det & Map & Motion | 0.41 | 0.70 | 1.05 | 0.72 | 0.07 | 0.17 | 0.41 | 0.22 | 0.60 | 2.38 | 5.18 | 2.72 | 5.3 |
| Drive-WM[‡] [38] | Det & Map | 0.43 | 0.77 | 1.20 | 0.80 | 0.10 | 0.21 | 0.48 | 0.26 | - | - | - | - | - |
| UAD[‡] (Ours) | None | 0.28 | 0.41 | 0.65 | 0.45 | 0.01 | 0.03 | 0.14 | 0.06 | 0.13 | 0.88 | 2.66 | 1.22 | 7.2 |
| UniAD[‡◊] [21] | Det&Track&Map&Motion&Occ | 0.20 | 0.42 | 0.75 | 0.46 | 0.02 | 0.25 | 0.84 | 0.37 | 0.20 | 1.33 | 3.24 | 1.59 | 2.1 |
| VAD-Base[‡◊] [22] | Det & Map & Motion | 0.17 | 0.34 | 0.60 | 0.37 | 0.04 | 0.27 | 0.67 | 0.33 | 0.21 | 2.13 | 5.06 | 2.47 | 5.3 |
| BEV-Planner[‡◊] [26] | None | 0.16 | 0.32 | 0.57 | 0.35 | 0.00 | 0.29 | 0.73 | 0.34 | 0.35 | 2.62 | 6.51 | 3.16 | - |
| UAD[‡◊] (Ours) | None | 0.13 | 0.28 | 0.48 | 0.30 | 0.00 | 0.12 | 0.55 | 0.22 | 0.10 | 0.80 | 2.48 | 1.13 | 7.2 |

## 4 Experiment

### 4.1 Experimental Setup

We conduct experiments in nuScenes [2] for open-loop evaluation, that contains 40,157 samples, of which 6,019 ones are used for evaluation. Following previous works [20, 21, 22], we adopt the metrics of L2 error (in meters) and collision rate (in percentage). Notably, the intersection rate with road boundary (in percentage), proposed in BEV-Planner [26], is also included for evaluation. For the closed-loop setting, we follow previous works [34, 20] to perform evaluation in the Town05 [34] benchmark of the CARLA simulator [11]. Route completion (in percentage) and driving score (in percentage) are used as the evaluation metrics. We adopt the query-based view transformer [25] to learn BEV features from multi-view images. The confidence threshold of the open-set 2D detector is set to 0.35 to filter unreliable predictions. The angle $\theta$ to partition the BEV space is set to $4°$ ($K=360°/4°$), and the default threshold $\delta$ is $1.2m$ (see Sec. 3.3). The weight coefficients in Eq. 8 are set to $2.0, 0.1, 1.0, 2.0, 1.0$. Our model is trained for 24 epochs on 8 NVIDIA Tesla A100 GPUs with a batch size of 1 per GPU. Other settings follow UniAD [21] unless otherwise specified.

We observed that ST-P3 [20] and VAD [22] adopt different open-loop evaluation protocols (L2 error and collision rate) from UniAD in their official codes. We denote the setting in ST-P3 and VAD as TemAvg and the one in UniAD as NoAvg, respectively. In specific, the TemAvg protocol calculates metrics by averaging the performances from 0.5s to the corresponding timestamp. Taking the L2 error at 2s as an example, the calculation in TemAvg is

$$L2@2s = \text{Avg}(l2_{0.5s}, l2_{1.0s}, l2_{1.5s}, l2_{2.0s}), \tag{9}$$

where Avg is the average operation and $0.5s$ is the time interval between two consecutive annotated frames in nuScenes [2]. For NoAvg protocol, $L2@2s = l2_{2.0s}$.

### 4.2 Comparison with State-of-the-arts

**Open-loop Evaluation.** Tab. 1 presents the performance comparison in terms of L2 error, collision rate, intersection rate with road boundary, and FPS. Since ST-P3 and VAD adopt different evaluation protocols from UniAD to compute L2 error and collision rate (see Sec. 4.1), we respectively calculate the results under different settings, *i.e.,* NoAvg and TemAvg. As shown in Tab. 1, the proposed UAD achieves superior planning performance over UniAD and VAD on all metrics, while running faster. Notably, our UAD obtains 39.4% and 55.2% relative improvements on Collision@3s compared with UniAD and VAD under the NoAvg evaluation protocol (*e.g.,* 39.4%=(0.71%-0.43%)/0.71%), demonstrating the longtime robustness of our method. Moreover, UAD runs at 7.2FPS, which is $3.4\times$ and $1.4\times$ faster than UniAD and VAD-Base, respectively, verifying the efficiency of our framework. Surprisingly, our tiny version, UAD-Tiny, which aligns the settings of backbone, image size, and BEV

Table 2: Closed-loop evaluation in the CARLA simulator [11]. † denotes the LiDAR-based method.

| Method | Town05 Short | | Town05 Long | |
|---|---|---|---|---|
| | Driving Score ↑ | Route Completion ↑ | Driving Score ↑ | Route Completion ↑ |
| CILRS [8] | 7.47 | 13.40 | 3.68 | 7.19 |
| LBC [5] | 30.97 | 55.01 | 7.05 | 32.09 |
| Transfuser† [34] | 54.52 | 78.41 | 33.15 | 56.36 |
| ST-P3 [20] | 55.14 | 86.74 | 11.45 | 83.15 |
| VAD-Base [22] | 64.29 | 87.26 | 30.31 | 75.20 |
| UAD (Ours) | **67.83** | **91.05** | **36.71** | **85.12** |

Table 3: Ablation on the loss functions. We evaluate the influence of each designed module by applying corresponding loss.

| # | $\mathcal{L}_{spat}$ | $\mathcal{L}_{drm}$ | $\mathcal{L}_{dir}$ | $\mathcal{L}_{cons}$ | $\mathcal{L}_{imi}$ | L2 (m) ↓ | | | | Collision (%) ↓ | | | |
|---|---|---|---|---|---|---|---|---|---|---|---|---|---|
| | | | | | | 1s | 2s | 3s | Avg. | 1s | 2s | 3s | Avg. |
| ① | - | - | - | - | ✓ | 1.20 | 3.04 | 5.30 | 3.18 | 0.83 | 1.33 | 5.13 | 2.43 |
| ② | ✓ | - | - | - | ✓ | 0.44 | 0.93 | 1.64 | 1.00 | 0.30 | 0.56 | 1.28 | 0.71 |
| ③ | - | ✓ | - | - | ✓ | 0.51 | 1.12 | 1.97 | 1.20 | 0.71 | 1.13 | 2.71 | 1.52 |
| ④ | - | - | ✓ | - | ✓ | 0.83 | 1.57 | 2.40 | 1.60 | 0.79 | 1.29 | 3.89 | 1.99 |
| ⑤ | - | - | - | ✓ | ✓ | 0.59 | 1.30 | 2.34 | 1.41 | 0.76 | 1.25 | 3.47 | 1.83 |
| ⑥ | ✓ | ✓ | ✓ | ✓ | ✓ | **0.39** | **0.81** | **1.50** | **0.90** | **0.01** | **0.12** | **0.43** | **0.19** |

Table 4: Ablation on the dreaming decoder.

| # | Circular Update | Dreaming Loss | L2 (m) ↓ | | | | Collision (%) ↓ | | | |
|---|---|---|---|---|---|---|---|---|---|---|
| | | | 1s | 2s | 3s | Avg. | 1s | 2s | 3s | Avg. |
| ① | - | - | 0.98 | 1.73 | 2.74 | 1.82 | 0.43 | 0.85 | 1.71 | 1.00 |
| ② | ✓ | - | 0.50 | 0.98 | 1.87 | 1.12 | 0.27 | 0.60 | 1.37 | 0.75 |
| ③ | - | ✓ | 0.44 | 0.96 | 1.73 | 1.04 | 0.08 | 0.35 | 1.13 | 0.52 |
| ④ | ✓ | ✓ | **0.39** | **0.81** | **1.50** | **0.90** | **0.01** | **0.12** | **0.43** | **0.19** |

Table 5: Ablation on direction-aware learning strategy.

| # | Directional Augment | Directional Consistency | L2 (m) ↓ | | | | Collision (%) ↓ | | | |
|---|---|---|---|---|---|---|---|---|---|---|
| | | | 1s | 2s | 3s | Avg. | 1s | 2s | 3s | Avg. |
| ① | - | - | 0.42 | 0.88 | 1.61 | 0.97 | 0.05 | 0.18 | 0.73 | 0.32 |
| ② | ✓ | - | 0.41 | 0.83 | 1.53 | 0.92 | 0.05 | 0.23 | 0.68 | 0.32 |
| ③ | ✓ | ✓ | **0.39** | **0.81** | **1.50** | **0.90** | **0.01** | **0.12** | **0.43** | **0.19** |

resolution in VAD-Tiny, runs at the fastest speed of 18.9FPS while clearly outperforming VAD-Tiny and even achieving comparable performance with VAD-Base. This again proves the superiority of our design. More detailed runtime comparisons and analyses are presented in the appendix. We adopt the NoAvg evaluation protocol in the following ablation experiments unless otherwise specified. Recent works discuss the effect of using ego status in the planning module [22, 26]. Following this trend, we also fairly compare the ego status equipped version of our model with these works. It shows that the superiority of our UAD is still preserved, which also achieves the best performance against the compared methods. Moreover, BEV-Planner [26] introduces a new metric named "interaction" for better evaluating the performance of E2EAD methods. As shown in Tab. 1, our model obtains the average interaction rate of 1.13%, obviously outperforming other methods. This again proves the effectiveness of our UAD. On the other hand, this demonstrates the importance of designing a suitable pretext for perceiving the environment. Only using ego status is not enough for safe driving.

**Closed-loop Evaluation.** The simulation results in CARLA [11] are shown in Tab. 2. Our UAD achieves better performance compared with recent E2E planners ST-P3 [20] and VAD [22] in all scenarios, proving the effectiveness. Notably, on challenging Town05 Long benchmark, UAD greatly outperforms recent E2E method VAD by 6.40 points on the driving score and 9.92 points on route completion, respectively. This proves the reliability of our UAD for long-term autonomous driving.

### 4.3 Component-wise Ablation

**Loss Functions.** We first analyze the influence of different loss functions that correspond to the proposed pretext task and self-supervised trajectory learning strategy. The experiments are conducted on the validation split of the nuScenes [2], as shown in Tab. 3. The model with single imitation loss $\mathcal{L}_{imi}$ is considered as the baseline (①). With the enhanced perception capability by the spatial objectness loss $\mathcal{L}_{spat}$, the average L2 error and collision rate are clearly improved to 1.00m and 0.71% from 3.18m and 2.43%, respectively (② *v.s.* ①). The dreaming loss $\mathcal{L}_{drm}$, direction loss $\mathcal{L}_{dir}$ and consistency loss $\mathcal{L}_{cons}$ also respectively bring considerable gains on the average L2 error for 1.98m, 1.58m, 1.77m over the baseline model (③,④,⑤ *v.s.* ①). The loss functions are finally combined to construct our UAD (⑥), which obtains the average L2 error of 0.90m and average collision rate of 0.19%. The results demonstrate the effectiveness of each proposed component.

**Temporal Learning with Dreaming Decoder.** The temporal learning with the proposed dreaming decoder is realized by Circular Update and Dreaming Loss. The circular update is in charge of both extracting information from observed scenes (Eq. 2) and generating pseudo observations to predict the ego trajectories of future frames (Eq. 3). We study the influence of each module in Tab. 4. Circular Update and Dreaming Loss respectively bring performance gains of 0.70m/0.78m on the average L2 error (②,③*v.s.*①), proving the effectiveness of our designs. Applying both two modules (④) achieves the best performance, showing their complementarity for temporal representation learning.

**Direction Aware Learning Strategy.** Directional Augmentation and Directional Consistency are the two core components of the proposed direction-aware learning strategy. We prove their effectiveness in Tab. 5. It shows that the Directional Augmentation improves the average L2 error for considerable

Table 7: Performances under different driving scenes. * denotes not using direction-aware learning.

| Method | Perf. *go straight* ↓ (5309 samples) | | Perf. *turn left* ↓ (301 samples) | | Perf. *turn right* ↓ (409 samples) | | Perf. *Overall* ↓ (6019 samples) | |
| --- | --- | --- | --- | --- | --- | --- | --- | --- |
| | Avg. L2 (m) | Avg. Col. (%) | Avg. L2 (m) | Avg. Col. (%) | Avg. L2 (m) | Avg. Col. (%) | Avg. L2 (m) | Avg. Col. (%) |
| UniAD [21] | 0.98 | 0.26 | 1.48 | 0.55 | 1.27 | 0.73 | 1.03 | 0.31 |
| VAD-Base [22] | 1.19 | 0.37 | 1.47 | 0.78 | 1.39 | 0.81 | 1.22 | 0.43 |
| UAD* (Ours) | 0.89 | 0.28 | 1.55 | 0.43 | 1.51 | 0.65 | 0.97 | 0.32 |
| UAD (Ours) | **0.84** | **0.17** | **1.39** | **0.22** | **1.16** | **0.33** | **0.90** | **0.19** |

0.05m (②v.s.①). One interesting observation is that applying the augmentation brings more gains for long-term planning than short-term ones, *i.e.,* the L2 error of 1s/3s decreases for 0.01m/0.08m compared with ①, which proves the effectiveness of our augmentation on enhancing longer temporal information. The Directional Consistency further reduces the average collision rate for impressive 0.13% (③v.s.②), which enhances the robustness for driving directional change.

**Angular Design.** We further explore the influence of the proposed angular design by removing the angular partition and angular queries. Specifically, the BEV feature is directly fed into the dreaming decoder to predict pixel-wise objectness, which is supervised by the BEV object mask (see Fig. 2) with binary cross-entropy loss. Besides, the ego query directly interacts with the BEV feature by cross-attention to extract environmental information. The results are presented in Tab. 6.

When discarding the angular design, the average L2 error degrades for 0.47m, and the average collision rate consistently degrades for 1.18%. This demonstrates the effectiveness of our angular design in perceiving complex environments and planning robust driving routes.

Table 6: Ablation on the angular design.

| # | Angular Design | L2 (m) ↓ | | | | Collision (%) ↓ | | | |
| --- | --- | --- | --- | --- | --- | --- | --- | --- | --- |
| | | 1s | 2s | 3s | Avg. | 1s | 2s | 3s | Avg. |
| ① | - | 0.78 | 1.31 | 2.01 | 1.37 | 0.61 | 1.39 | 2.12 | 1.37 |
| ② | ✓ | **0.39** | **0.81** | **1.50** | **0.90** | **0.01** | **0.12** | **0.43** | **0.19** |

### 4.4 Further Analysis

**Planning Performance in Different Driving Scenes.** The direction-aware learning strategy is designed to enhance the planning performance in scenarios of vehicle steering. We demonstrate the superiority of our proposed model by evaluating the metrics of different driving scenes in Tab. 7. According to the given driving command (*i.e., go straight*, *turn left* and *turn right*), we divide the 6,019 validation samples in nuScenes [2] into three parts, which contain 5,309, 301 and 409 ones, respectively. Not surprisingly, all methods perform better under *go straight* scenes than the steering scenes, proving the necessity of augmenting the imbalanced training data for robust planning. When applying the proposed direction-aware learning strategy, our UAD achieves considerable gains on the average collision rate of *turn left* and *turn right* scenes (UAD *v.s.* UAD*). Notably, our model outperforms UniAD and VAD by a large margin in steering scenes, proving its effectiveness.

**Visualization of Angular Perception and Planning.** The angular perception pretext is designed to perceive the objects in each sector region. We show its capability by visualizing the predicted objectness in nuScenes [2] in Fig. 5a. For a better view, we transform the discrete objectness scores and ground truth to a pseudo-BEV mask. It shows that our model can successfully capture surrounding objects. Fig. 5a also shows the open-loop planning results of recent SOTA UniAD [21], VAD [22] and our UAD, proving the effectiveness of our method to plan a more reasonable ego trajectory. Fig. 5b compares the closed-loop driving routes between Transfuser [34], ST-P3 [20] and our UAD in CARLA [11]. Our method successfully notices the person and drives in a much safer manner, proving the reliability of our UAD in handling safe-critical issues under complex scenarios.

Due to limited space, we present more analyses in the appendix, including **1)** the influence of partition angle $\theta$, **2)** the influence of direction threshold $\delta$, **3)** different backbones and pre-trained weights, **4)** replacing 2D ROIs from GroundingDINO with 2D GT boxes, **5)** different settings of GroundingDINO to generate 2D ROIs, **6)** the influence of pre-training to previous method UniAD and our UAD, **7)** runtime analysis of each module in our UAD and modularized UniAD, **8)** more visualizations, *etc*.

### 4.5 Discussion

**Ego Status and Open-loop Planning Evaluation.** As revealed by [26, 40], it's not a challenge to acquire decent performance of L2 error and collision rate (the original metrics in nuScenes [2]) in the open-loop evaluation of nuScenes by using ego status in the planning module (see Tab. 1). The question is: *is open-loop evaluation meaningless?* Our answer is **NO**. Firstly, the inherent reason for the observation is that the simple cases of *go straight* dominate the nuScenes testing dataset. In these

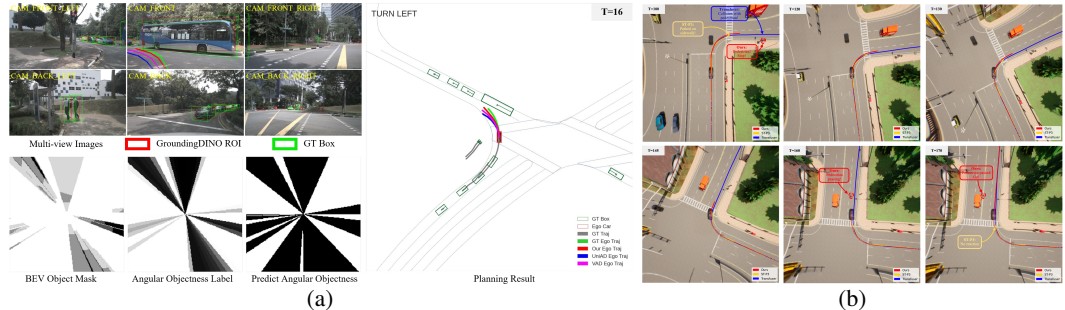

Figure 5: **(a)** Qualitative results in nuScenes. **(b)** Qualitative results in CARLA.

cases, even a linear extrapolation of motion being sufficient for planning is not surprising. However, as shown in Tab. 7, in more challenging cases like *turn right* and *turn left*, the open-loop metrics can still clearly indicate the difficulty of steering scenarios and the differences in methods, which is also proved in [26]. Therefore, open-loop evaluation is not meaningless, while the crux is the distribution of the testing data and the metrics. Secondly, the advantage of open-loop evaluation is its efficiency, which benefits the fast development of algorithms. This view is also revealed by a recent simulator design study [9], which tries to transform the closed-loop evaluation into an open-loop fashion.

In our work, we thoroughly compare our model with other methods, which shows consistent improvements against previous works under various driving scenarios (straight or steering), different usage of ego status (*w/.* or *w/o.*), diverse evaluation metrics (L2 error, collision rate or intersection rate from [26]), and different evaluation types (open- or closed-loop). It thus again proves the importance of designing suitable pretext tasks for end-to-end autonomous driving.

**How to Guarantee Safety in Current Auto-Drive System?** Safety is the first requirement of autonomous driving systems in practical products, especially for L4-level auto-vehicles. To guarantee safety, offline collision check with predicted 3D boxes is an inevitable post-process under current technological conditions. Then, a question naturally arises: *how to safely apply our model to current auto-driving systems?* Before answering this question, we reaffirm our claim that we believe discarding 3D labels is an efficient, attractive, and potential direction for E2EAD, but it doesn't mean we refuse to use any 3D labels if the relatively cheap ones are available in practical product engineering. For instance, solely annotating bounding boxes without object identity for tracking is much cheaper than labeling other elements like HD-map, and point-cloud segmentation labels for occupancy. Therefore, we provide a degraded version of our method by arranging an additional 3D detection head. Then our model can seamlessly integrate into auto-drive products, and offline collision check is achievable. As shown in Tab. 8, integrating the 3D detection head doesn't bring additional improvements, which again proves the design of our method has sufficiently encoded 3D information to the planning module.

Table 8: Ablation on the 3D detection head.

| # | Detection Head | L2 (m) ↓ | | | | Collision (%) ↓ | | | |
|---|---|---|---|---|---|---|---|---|---|
| | | 1s | 2s | 3s | Avg. | 1s | 2s | 3s | Avg. |
| ① | - | **0.39** | **0.81** | **1.50** | **0.90** | **0.01** | **0.12** | **0.43** | **0.19** |
| ② | ✓ | 0.37 | 0.86 | 1.57 | 0.93 | 0.02 | 0.17 | 0.55 | 0.25 |

In a nutshell, **1)** our work can easily integrate other 3D tasks if they are inevitable under current technical conditions; **2)** the experiments again prove from the side that our spatial-temporal module has already encoded important 3D clues for planning; **3)** we hope our frontier work can eliminate some inessential 3D sub-tasks for both research and engineer usage of E2EAD models. An era of cheap, laboratory-affordable but robust, practical E2EAD design will eventually come!

## 5 Conclusion

Our work seeks to liberate E2EAD from costly modularization and 3D manual annotation. With this goal, we propose the unsupervised pretext task to perceive the environment by predicting angular-wise objectness and future dynamics. To improve the robustness in steering scenarios, we introduce the direction-aware training strategy for planning. Experiments demonstrate the effectiveness and efficiency of our method. As discussed, although the ego trajectories are easily obtained, it is almost impossible to collect billion-level precisely annotated data with perception labels. This impedes the further development of end-to-end autonomous driving. We believe our work provides a potential solution to this barrier and may push performance to the next level when massive data are available.

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

# A Appendix

The appendix presents additional designing and explaining details of our **U**npervised pretext task for end-to-end **A**utonomous **D**riving (UAD) in the manuscript.

- **Different Partition Angles**
  We explore the influence of different partition angles in angular pretext to learn better spatio-temporal knowledge.

- **Different Direction Thresholds**
  We explore the influence of different thresholds in direction prediction to enhance planning robustness in complex driving scenarios.

- **Different Backbones and Pre-trained Weights**
  We compare the performance of different backbones and pre-trained weights on our method.

- **Objectness Label Generation with GT Boxes**
  We compare the generated objectness label between using the pseudo ROIs from GroundingDINO [28] and ground-truth boxes on different backbones.

- **Settings for ROI Generation**
  We ablate different settings for the open-set 2D detector GroundingDINO, which provides ROIs for the label generation of angular perception pretext.

- **Different Image Sizes and BEV Resolution**
  We compare the performance with different input sizes of multi-view images and BEV resolutions.

- **Runtime Analysis**
  We evaluate the runtime of each module of UAD and compare with modularized UniAD [21], which demonstrates the efficiency of our method.

- **Classification of Angular Perception**
  We evaluate the objectness prediction in the angular perception pretext, which demonstrates the enhanced perception capability in complex driving scenarios.

- **Influence of Pre-training**
  We evaluate the influence of pre-training by detailing the training losses and planning performances with different pre-trained weights.

- **More Visualizations**
  We provide more visualizations for the predicted angular-wise objectness and planning results in the open-loop evaluation of nuScenes [2] and closed-loop simulation of CARLA [11].

## A.1 Different Partition Angles

The proposed angular perception pretext divides the BEV space into multiple sectors. We explore the influence of partition angle $\theta$ in Tab 9. Experimental results show that the L2 error and inference speed gradually increase with the partition angle. The model with partition angle of $1°$(①) achieves the best average L2 error of 0.85m. And the partition angle of $4°$ contributes to the best average collision rate of 0.19% (③). This reveals that a smaller partition angle helps learn more fine-grained environmental representations, eventually benefiting planning. In contrast, the model with a large partition angle sparsely perceives the scene. Despite reducing the computation cost, it will also degrade the safety of the end-to-end autonomous driving system.

Table 9: Ablation on different partition angles in the proposed angular pretext.

| # | Partition Angle | L2 (m) ↓ | | | | Collision (%) ↓ | | | | FPS |
|---|---|---|---|---|---|---|---|---|---|---|
| | | 1s | 2s | 3s | Avg. | 1s | 2s | 3s | Avg. | |
| ① | 1° | 0.35 | 0.78 | **1.42** | **0.85** | 0.01 | 0.28 | 0.68 | 0.32 | 5.0 |
| ② | 2° | **0.34** | **0.77** | 1.46 | 0.86 | 0.01 | 0.22 | 0.48 | 0.24 | 6.3 |
| ③ | 4° | 0.39 | 0.81 | 1.50 | 0.90 | **0.01** | **0.12** | **0.43** | **0.19** | 7.2 |
| ④ | 8° | 0.38 | 0.85 | 1.55 | 0.93 | 0.01 | 0.18 | 0.55 | 0.25 | 7.7 |
| ⑤ | 15° | 0.47 | 0.94 | 1.69 | 1.03 | 0.03 | 0.20 | 0.60 | 0.28 | 8.1 |
| ⑥ | 30° | 0.48 | 1.00 | 1.75 | 1.08 | 0.05 | 0.28 | 0.63 | 0.32 | **8.4** |

Table 10: Ablation on different thresholds of direction prediction in the directional augmentation.

| # | Threshold (m) | L2 (m) ↓ | | | | Collision (%) ↓ | | | |
|---|---|---|---|---|---|---|---|---|---|
| | | 1s | 2s | 3s | Avg. | 1s | 2s | 3s | Avg. |
| ① | 0.5 | **0.35** | 0.79 | **1.43** | **0.86** | 0.03 | 0.18 | 0.71 | 0.31 |
| ② | 0.8 | 0.35 | **0.77** | 1.46 | 0.86 | 0.01 | 0.12 | 0.68 | 0.27 |
| ③ | 1.2 | 0.39 | 0.81 | 1.50 | 0.90 | 0.01 | 0.12 | 0.43 | **0.19** |
| ④ | 1.5 | 0.40 | 0.82 | 1.52 | 0.91 | 0.02 | 0.15 | **0.42** | 0.20 |
| ⑤ | 2.0 | 0.38 | 0.85 | 1.55 | 0.93 | **0.01** | **0.08** | 0.48 | 0.19 |

Table 11: Ablation on different backbones and pre-trained weights.

| # | Backbone | Pretrained Weight | L2 (m) ↓ | | | | Collision (%) ↓ | | | | FPS |
|---|---|---|---|---|---|---|---|---|---|---|---|
| | | | 1s | 2s | 3s | Avg. | 1s | 2s | 3s | Avg. | |
| ① | Res50 | None | 0.43 | 0.94 | 1.65 | 1.01 | 0.03 | 0.37 | 0.86 | 0.42 | 9.6 |
| ② | | ImageNet | 0.41 | 0.90 | 1.66 | 0.99 | 0.03 | 0.32 | 0.80 | 0.38 | |
| ③ | Res101 | None | 0.40 | 0.87 | 1.59 | 0.95 | 0.02 | 0.23 | 0.59 | 0.28 | 7.2 |
| ④ | | ImageNet | 0.37 | 0.84 | 1.53 | 0.91 | 0.01 | 0.18 | 0.50 | 0.23 | |
| ⑤ | | COCO | **0.36** | 0.83 | 1.51 | 0.90 | 0.01 | 0.16 | 0.45 | 0.21 | |
| ⑥ | | NuImages | 0.39 | **0.81** | **1.50** | **0.90** | **0.01** | **0.12** | **0.43** | **0.19** | |

Table 12: Ablation on 2D object boxes in pretext label generation.

| # | Backbone | 2D Object Box | L2 (m) ↓ | | | | Collision (%) ↓ | | | | FPS |
|---|---|---|---|---|---|---|---|---|---|---|---|
| | | | 1s | 2s | 3s | Avg. | 1s | 2s | 3s | Avg. | |
| ① | Res50 | Pseudo | 0.41 | 0.90 | 1.66 | 0.99 | 0.03 | 0.32 | 0.80 | 0.38 | 9.6 |
| ② | | GT | 0.41 | 0.87 | 1.61 | 0.96 | 0.03 | 0.30 | 0.71 | 0.35 | |
| ③ | Res101 | Pseudo | 0.39 | 0.81 | 1.50 | 0.90 | 0.01 | **0.12** | 0.43 | 0.19 | 7.2 |
| ④ | | GT | **0.37** | **0.79** | **1.45** | **0.84** | **0.01** | 0.13 | **0.39** | **0.18** | |

## A.2 Different Direction Thresholds

The direction prediction that the ego car intends to maneuver (*i.e., left*, *straight* and *right*) is proposed to enhance the steering capability for autonomous driving. The label is generated with the threshold $\delta$ (see Eq. 7 in the manuscript), which determines the ground-truth direction of each waypoint in the expert trajectory. Here we explore the influence by ablating different thresholds, as shown in Tab. 10. Experimental results show that the L2 error gradually increases with the direction threshold. The model with $\delta$ of 0.5m (①) achieves the lowest L2 error of 0.86m. It reveals that a smaller threshold will force the planner to fit the expert navigation, leading to a closer distance between the predicted trajectory and the ground truth. In contrast, the collision rate benefits more from larger thresholds. The model with $\delta$ of 2.0m obtains the best collision rate at 2s of 0.08% (⑤), showing the effectiveness for robust planning. Notably, the threshold of 1.2m contributes to a great balance with the average L2 error of 0.90m and average collision rate of 0.19%.

## A.3 Different Backbones and Pre-trained Weights

As a common sense, pre-training the backbone network with fundamental tasks like image classification on ImageNet [10] will benefit the sub-tasks. The previous method UniAD [21] uses the pre-trained weights of BEVFormer [25]. What surprised us is that when replacing the pre-trained weights with the one learned on ImageNet, the performance of UniAD dramatically degraded (see "Influence of Pre-training" for more details). This inspires us to explore the influence of backbone settings on our framework. As shown in Tab. 11, interestingly, even without any pre-training, our model still outperforms UniAD with pre-trained ResNet101 and VAD with pre-trained ResNet50. This verifies the effectiveness of our unsupervised pretext task on modeling the driving scenes. We also use publicly available pre-trained weights on detection datasets like COCO [27] and nuImages [2] to train our model, which shows better performance. These experimental results and observations demonstrate that a potentially promising topic is *how to pre-train a model for end-to-end autonomous driving*. We leave this to future research.

## A.4 Objectness Label Generation with GT Boxes

As mentioned in the manuscript, the essence of generating the angular objectness label lies in the 2D ROIs, which come from the open-set 2D detector GroundingDINO [28]. Here we explore the influence of using the ground-truth 2D boxes as ROIs, which provide more high-quality samples for the representation learning in the angular perception pretext. Tab. 12 shows that training with GT boxes achieves consistent performance gains on both ResNet50 [17] and ResNet101 [17] (②,④ *v.s.* ①,③). This reveals that accurate annotation does help to learn better spatio-temporal knowledge and improve ego planning. Considering the cost in real-world deployment, training with accessible

Table 13: Ablation on the settings of ROI generation. The Conf. Thresh denotes the confidence threshold in GroundingDINO [28] to filter unreliable predictions. *vehicle,pedestrian,barrier* represent the used prompt words to obtain ROIs of corresponding classes. Rule Filter indicates filtering the ROIs that are more than half of the length or width of the image.

| # | Conf. Thresh | Prompt Words | Rule Filter | L2 (m) ↓ 1s | 2s | 3s | Avg. | Collision (%) ↓ 1s | 2s | 3s | Avg. |
|---|---|---|---|---|---|---|---|---|---|---|---|
| ① | 0.35 | {*vehicle*} | - | 0.48 | 0.98 | 1.75 | 1.07 | 0.08 | 0.38 | 0.80 | 0.42 |
| ② | 0.35 | {*vehicle,pedestrian*} | - | 0.47 | 0.94 | 1.69 | 1.03 | 0.04 | 0.27 | 0.71 | 0.34 |
| ③ | 0.35 | {*vehicle,pedestrian,barrier*} | - | 0.43 | 0.88 | 1.60 | 0.97 | 0.03 | 0.23 | 0.60 | 0.29 |
| ④ | 0.35 | {*vehicle,pedestrian,barrier*} | ✓ | **0.39** | **0.81** | 1.50 | 0.90 | **0.01** | **0.12** | 0.43 | 0.19 |
| ⑤ | 0.30 | {*vehicle,pedestrian,barrier*} | ✓ | 0.39 | 0.82 | **1.45** | **0.89** | 0.01 | 0.21 | 0.51 | 0.24 |
| ⑥ | 0.40 | {*vehicle,pedestrian,barrier*} | ✓ | 0.46 | 0.90 | 1.57 | 0.98 | 0.01 | 0.13 | **0.37** | **0.17** |

Table 14: Comparison with different backbones, image sizes and BEV resolutions.

| # | Method | Backbone | Image Size | BEV Resolution | L2 (m) ↓ 1s | 2s | 3s | Avg. | Collision (%) ↓ 1s | 2s | 3s | Avg. | FPS |
|---|---|---|---|---|---|---|---|---|---|---|---|---|---|
| ① | UniAD [21] | R101 | 1600×900 | 200×200 | 0.48 | 0.96 | 1.65 | 1.03 | 0.05 | 0.17 | 0.71 | 0.31 | 2.1 |
| ② | VAD-Tiny [22] | R50 | 640×360 | 100×100 | 0.60 | 1.23 | 2.06 | 1.30 | 0.33 | 1.33 | 2.21 | 1.29 | 17.6 |
| ③ | VAD-Base [22] | R50 | 1280×720 | 200×200 | 0.54 | 1.15 | 1.98 | 1.22 | 0.10 | 0.24 | 0.96 | 0.43 | 5.3 |
| ④ | UAD (Ours) | R50 | 640×360 | 100×100 | 0.47 | 0.99 | 1.71 | 1.06 | 0.08 | 0.39 | 0.90 | 0.46 | 18.9 |
| ⑤ | UAD (Ours) | R50 | 1600×900 | 200×200 | 0.41 | 0.90 | 1.66 | 0.99 | 0.03 | 0.32 | 0.80 | 0.38 | 9.6 |
| ⑥ | UAD (Ours) | R101 | 1600×900 | 200×200 | **0.39** | **0.81** | **1.50** | **0.90** | **0.01** | **0.12** | **0.43** | **0.19** | 7.2 |

pseudo labels is a more efficient way compared with the manual annotation, which also shows comparable performance in autonomous driving (① *v.s.* ② and ③ *v.s.* ④).

## A.5 Settings for ROI Generation.

The quality of learned spatio-temporal knowledge highly relies on the generated ROIs by the open-set 2D detector GroundingDINO [28], which are then projected as the BEV objectness label for training the angular perception pretext. We explore the influence of generated ROIs with different settings, as shown in Tab. 13. We take the setting with the confidence score of 0.35, prompt word of *vehicle* and without the Rule Filter, as the baseline (①). By appending more prompt words (*e.g., pedestrian*, *barrier*), the planning performance gradually improves (③,② *v.s.*①), showing the enhanced perception capability with more diversified objects. Filtering the ROIs with overlarge size (*i.e.,* Rule Filter) brings considerable gains for the average L2 error of 0.07m and average collision rate of 0.10% (④*v.s.*③). One interesting observation is that decreasing the confidence threshold would slightly improve the L2 error while causing higher collision rate (⑤*v.s.*④). In contrast, increasing the threshold obtains lower average collision rate of 0.17% and higher average L2 error of 0.98m. This reveals the importance of providing diversified ROIs for angular perception learning as well as ensuring high quality. The model with the confidence score of 0.35, all prompt words and Rule Filter achieves balanced performance with the average L2 error of 0.90m and average collision rate of 0.19%.

## A.6 Different Image Sizes and BEV Resolution

For safe autonomous driving, increasing the input size of the multi-view images and the resolution of the built BEV representation is an effective way, which provide more detailed environmental information. While benefiting perception and planning, it inevitably brings heavy computation cost. We then ablate the image size and BEV resolution of our UAD to find a balanced version between performance and efficiency, as shown in Tab. 14. The results show that our UAD with ResNet-101 [17], image size of 1600×900, BEV resolution of 200×200, achieves the best performance compared with previous methods UniAD [21] and VAD-Base [22] while running faster with 7.2FPS (⑥). By replacing the backbone with ResNet-50, our UAD is more efficient with little performance degradation (⑤ *v.s.* ⑥). We further align the settings of VAD-Tiny, which has an inference speed of outstanding 17.6FPS (②), to explore the influence of much smaller input sizes. Tab. 14 shows that our UAD still achieves excellent performance even compared with VAD-Base of high-resolution inputs (④ *v.s.* ③). Notably, our UAD of this version has the fastest inference speed of 18.9FPS. This

Table 15: Module runtime comparison between UniAD [21] and our UAD. The inference is measured on an NVIDIA Tesla A100 GPU.

| Model Partition | UniAD | | | UAD (Ours) | | |
|---|---|---|---|---|---|---|
| | Module | Latency (ms) | Proportion (%) | Module | Latency (ms) | Proportion (%) |
| Feature Extraction | Backbone | 38.1±0.5 | 8.2% | Backbone | 36.0±0.3 | 26.0% |
| | BEV Encoder | 83.4±0.5 | 17.9% | BEV Encoder | 81.5±0.4 | 58.9% |
| Sub-Task | Det&Track | 145.3±1.3 | 31.2% | Angular Partition | 1.1±0.1 | 0.8% |
| | Map | 92.1±0.7 | 19.8% | Dreaming Decoder | 18.2±0.2 | 13.2% |
| | Motion | 50.6±0.6 | 10.9% | | | |
| | Occupancy | 45.9±0.4 | 9.9% | | | |
| Prediction | Planning Head | 9.7±0.3 | 2.1% | Planning Head | 1.5±0.1 | 1.1% |
| Total | - | 465.1±4.3 | 100% | - | 138.3±1.1 | 100.0% |

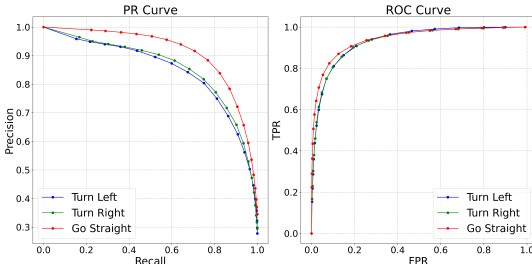

Figure 6: Visualization of the PR and ROC curves for the angular-wise objectness prediction in different driving scenes.

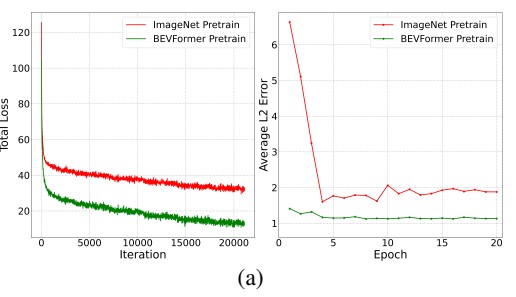
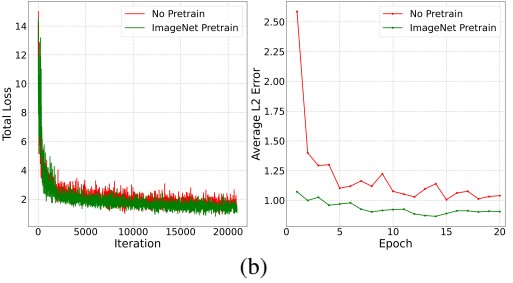

(a)          (b)

Figure 7: Optimization of UniAD **(a)** and our UAD **(b)** with different pre-trained backbone weights.

again proves the effectiveness of our method in performing fine-grained perception, as well as the robustness to fit the inputs of different sizes.

## A.7 Runtime Analysis

Tab. 15 compares the runtime of each module between the modularized method UniAD [21] and our UAD. As we adopt the Backbone and BEV Encoder from BEVFormer [25] that are the same in UniAD, the latency of feature extraction is similar with little difference due to different pre-processing. The modular sub-tasks in UniAD consume most of the runtime, *i.e.,* significant 71.8% for Det&Track (31.2%), Map (19.8%), Motion (10.9%) and Occupancy (9.9%), respectively. In contrast, our UAD performs simple Angular Partition and Dreaming Decoder, which take only 14.0% (19.3ms) to model the complex environment. This demonstrates our insight that it's a necessity to liberate end-to-end autonomous driving from costly modularization. The downstream Planning Head takes negligible 1.5ms to plan the ego trajectory, compared with 9.7ms in UniAD. Finally, our UAD finishes the inference with a total runtime of 138.3ms, 3.4× faster than the 465.1ms of UniAD, showing the efficiency of our design.

## A.8 Classification of Angular Perception

The proposed angular perception pretext learns spatio-temporal knowledge of the driving scene by predicting the objectness of each sector region, which is supervised by the generated binary angular-wise label. We show the perception ability by evaluating the classification metrics based on the validation split of the nuScenes [2] dataset. Fig. 6 draws the Precision-Recall (PR) curve and Receiver-Operating-Characteristic (ROC) curve in different driving scenes (*i.e., turn left*, *go straight* and *turn right*). In the PR curve, our UAD achieves balanced precision and recall scores in different driving scenes, showing the effectiveness of our pretext task to perceive the surrounding objects. Notably, the performance of *go straight* scenes is slightly better than the steering ones under all thresholds. This proves our insight to design tailored direction-aware learning strategy for improving the safety-critical *turn left* and *turn right* scenes. The ROC curve shows the robustness of our angular perception pretext to classify the objects from complex environmental observations.

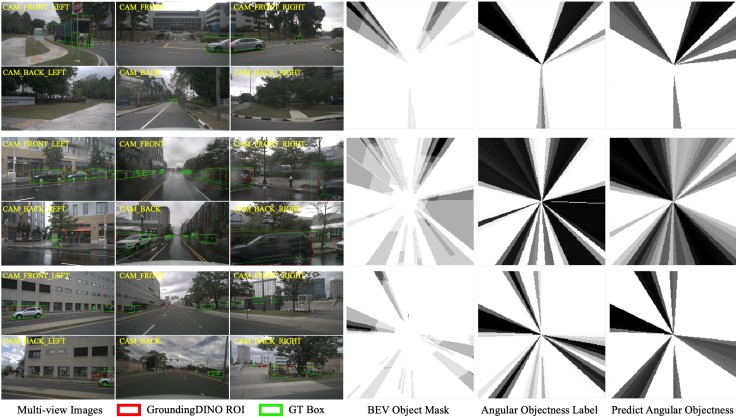

Figure 8: Visualization of the angular perception.

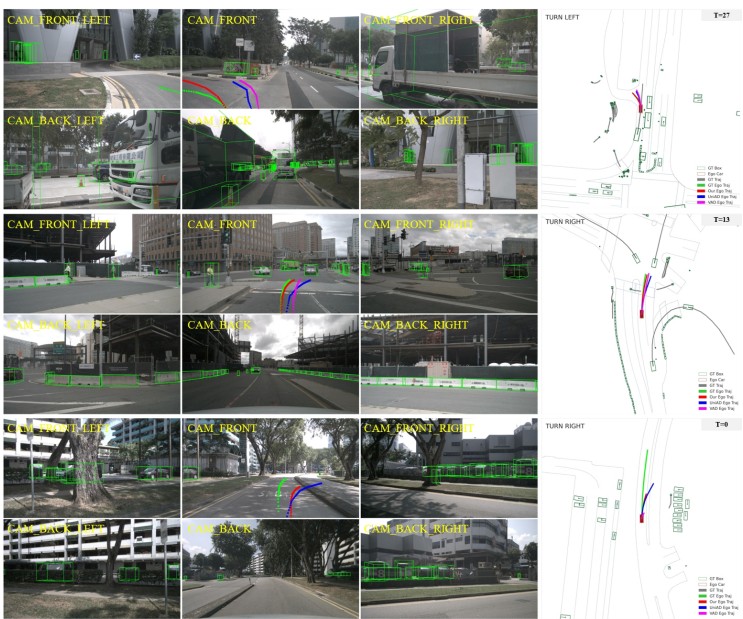

Figure 9: Visualization of the planning results. The first two rows show the success of our method in safe planning in complex scenarios, while the third row exhibits a failure case of our planner when no temporal information could be acquired when $t = 0$.

## A.9  Influence of Pre-training

Pre-training the backbone network with fundamental tasks is a commonly used metric to benefit representation learning. As mentioned in "Different Backbones and Pre-trained Weights" of Sec. 4.4 in the manuscript, the performance of the previous SOTA method UniAD [21] dramatically degrades without the pre-trained weights from BEVFormer [25]. Here we further detail the influence by comparing the training losses and planning performances with different pre-trained weights in Fig. 7. Fig. 7a shows that the training losses increase by about 20 on average when replaced with the pre-trained weights from ImageNet [10]. Correspondingly, the average L2 error is significantly higher than the one with the pre-trained weights from BEVFormer. This reveals that UniAD heavily relies on the perceptive pre-training in BEVFormer to optimize modularized sub-tasks. In contrast, our UAD performs comparably even without any pre-training (see Fig. 7b), proving the effectiveness of our designs for robust optimization.

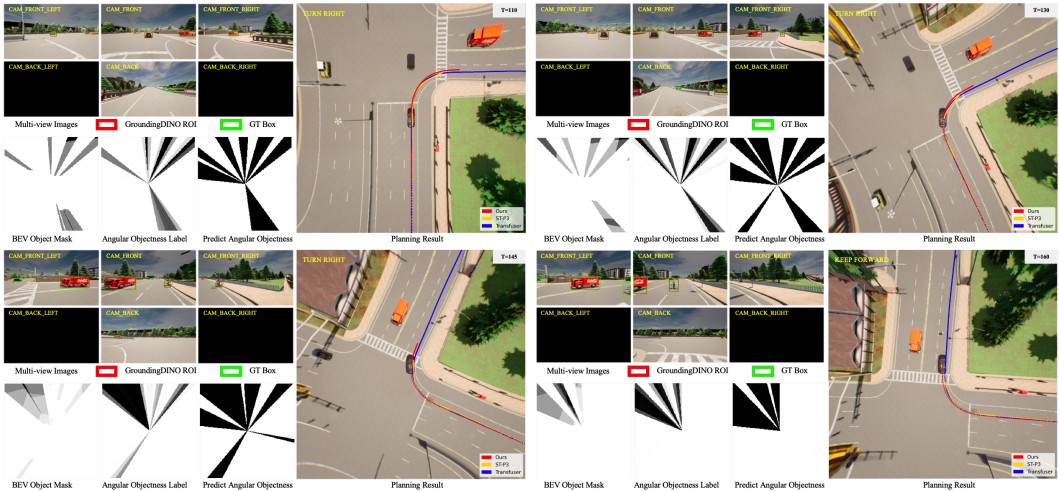

Figure 10: Visualization of angular perception and planning in Carla.

## A.10 More Visualizations

**Open-loop Planning**    We provide more visualizations about the predicted angular-wise objectness and planning results on nuScenes [2]. Fig. 8 compares the discrete objectness scores and ground truth, proving the effectiveness of our angular perception pretext to perceive the objects in each sector region. The planning results of previous SOTA methods (*i.e.,* UniAD [21] and VAD [22]) and our UAD are shown in Fig. 9. With the designed pretext and tailored training strategy, our method could plan a more reasonable ego trajectory under different driving scenarios, proving the effectiveness of our work. The third row shows the failure case of our planner. In this case, the ego car is given the "*Turn Right*" command when $t = 0$ (*i.e.,* the first frame of the driving scenario), leading to ineffectiveness of our planner in learning helpful temporal information. A possible solution to deal with this is to apply an auxiliary trajectory prior for the first several frames, and we leave this to future work.

**Closed-loop Simulation**    Fig. 10 visualizes the predicted objectness and planning results in the Town05 Long benchmark of CARLA [11]. Following the setting of ST-P3 [20] in closed-loop evaluation, we collect visual observations from the cameras of "CAM_FRONT", "CAM_FRONT_LEFT", "CAM_FRONT_RIGHT" and "CAM_BACK". It shows that the sector regions in which the surrounding objects exist are successfully captured by our UAD, proving the effectiveness and robustness of our design. Notably, the missed objects by GroundingDINO [28], *e.g.,* the black car in the camera of "CAM_FRONT_LEFT" at $t = 145$, are surprisingly perceived and marked in the corresponding sector. This demonstrates our method has the capability of learning perceptive knowledge in a data-driven manner, even with coarse supervision by the generated 2D pseudo boxes from GroundingDINO.

