# OpenReview forum: "End-to-End Autonomous Driving without Costly Modularization and 3D Manual Annotation"
_NeurIPS.cc/2024/Conference — Submitted to NeurIPS 2024_

### Official Review · Reviewer_YXrs · 2024-07-08

**Soundness:** 4
**Presentation:** 4
**Contribution:** 4
**Rating:** 9
**Confidence:** 5

**Summary:**

This paper handles the costly modularization and 3D manual annotation in current end-to-end autonomous driving, which proposes an unsupervised pretext task to provide necessary environmental information, as well as a direction-aware training strategy to enhance the robustness in safety-critical steering scenarios.

The authors conduct comprehensive experiments in both open- and closed-loop evaluation benchmarks, which demonstrate the effectiveness in various metrics. Moreover, the improvements are obtained with much less resource cost and faster inference speed, which is surprising and impressive.

In addition, this paper gives in-depth discussion and performance comparison about the usage of ego status in the open-loop evaluation of nuScenes. The considerable improvement in the intersection rate with the road boundary, which is proposed in recent BEV-Planner, again proves the superiority of the designed pretext task.

**Strengths:**

Overall, I am rather positive on this paper. In particular, I really like the motivation of this work that aims at finding a solution to relieve the heavy annotation and computation overload in current end-to-end autonomous driving. I believe this paper can inspire other works and facilitate this field. The strengths in this work include:

(1)	Enough novelty. This paper introduces an innovative unsupervised pretext task to perceive the environment, which is completely different from other works that accumulate subtasks requiring massive 3D annotation and computation resources.

(2)	Good performance. This paper demonstrates excellent performance and fast inference speed in both open- and closed-loop evaluation compared with other end-to-end methods. In specific on the challenging metric, i.e., intersection rate in BEV-Planner, the proposed approach surpasses other methods by a considerable margin. This clearly shows the effectiveness and advantages of the proposed method.

(3)	Insightful analysis. The authors provide extensive experiments and analysis for the proposed method. I appreciate this. The experimental analysis with various ablation studies allows a better understanding of each module. Notably, the authors observe the different computation ways of open-loop evaluation metrics between ST-P3 and UniAD and provide performance comparison with different settings, showing the comprehensiveness.

(4)	Good writing and organization. This paper is well-written and organized. Each section has a clear motivation. It’s easy to follow the ideas. I enjoy reading the paper.

Overall, I believe this paper is significant to the autonomous driving community because it shows new insights and directions in designing simple but effective E2EAD framework with SOTA performance.

**Weaknesses:**

(1) In this work, the 2D ROIs are crucial for the designed pretext task. I noticed that the authors adopt the open-set 2D detector GroundingDINO to generate the ROIs. Then the results and discussion of using other third-party detectors should be presented.
(2) The proposed method is shown to be efficient with the unsupervised pretext task and self-supervised training strategy, which is nice. It is suggested the authors show the influence of the training data volume (e.g., 25% and 50%).

**Questions:**

In the appendix, the authors show that the proposed method still achieves comparative performance even without backbone pretraining, while UniAD dramatically degrades without the pretrained weights of BEVFormer. What do you think are the reasons causing this?

**Limitations:**

It is suggested to provide discussion of limitations and broader impact in the revision.

---

> ### Author Rebuttal · Authors · 2024-08-06
>
> ## Response to Reviewer YXrs:
> We appreciate your careful review and thoughtful comments. We are encouraged and grateful that the reviewer found our approach to be well-motivated and insightful. Below, we address the concerns that were raised, and remain committed to clarifying further questions that may arise during the discussion period.
>
> ***
> ***Q1: In this work, the 2D ROIs are crucial for the designed pretext task. I noticed that the authors adopt the open-set 2D detector GroundingDINO to generate the ROIs. Then the results and discussion of using other third-party detectors should be presented.***
>
> **A1**: Thanks for the constructive comment. As suggested, we have evaluated different methods for generating 2D ROIs, including two open-set 2D detectors (GroundingDINO and DE-ViT[1]) and a 3D detector MV2D[2] (where the 3D predictions are projected onto the camera plane to obtain 2D ROIs). As shown in #Rebuttal-PDF-Fig.3(a), using ROIs from GroundingDINO results in the best performance. This highlights GroundingDINO's effectiveness in generating high-quality 2D ROIs compared to DE-ViT and MV2D.
>
> For further investigation, we assessed the 2D detection performance on nuScenes by matching the 2D ROIs with projected ground-truth 3D boxes. The PR and ROC curves are presented in #Rebuttal-PDF-Fig.3(b). GroundingDINO consistently outperforms the others, demonstrating its capability to perceive objects effectively in challenging driving scenarios. Notably, the 3D detector MV2D, which is trained on nuScenes data, does not show superior performance in either planning or 2D detection tasks. This suggests that current open-set 2D detectors, such as GroundingDINO, have significant potential for auto-annotating with prompt information and enhancing downstream tasks.
>
> We will include these results and analyses in the revision. Thanks.
>
> [1] Zhang X, et al. Detect every thing with few examples. In arXiv 2023.
>
> [2] Wang Z, et al. Object as query: Lifting any 2d object detector to 3d detection. In ICCV 2023.
>
> ***
> ***Q2: The proposed method is shown to be efficient with the unsupervised pretext task and self-supervised training strategy, which is nice. It is suggested the authors show the influence of the training data volume (e.g., 25% and 50%).***
>
> **A2**: Thanks for this thoughtful comment. As suggested, we trained our UAD model with different volumes of training data, and the results are presented in #Rebuttal-PDF-Tab.4. The comparison shows that even with only 50% of the data, our UAD outperforms the baseline models UniAD and VAD. Increasing the amount of data further enhances the model's performance. It is also observed that more challenging steering scenarios, such as turning right and left, offer greater potential for improvement. We believe that the performance in these challenging scenarios could reach new heights with the availability of more data. This will be a focus of our future research.
>
> We will include the analysis and results in the revision. Thanks, again.
>
> ***
> ***Q3: In the appendix, the authors show that the proposed method still achieves comparative performance even without backbone pretraining, while UniAD dramatically degrades without the pretrained weights of BEVFormer. What do you think are the reasons causing this?***
>
> **A3**: Thanks for this insightful comment. As discussed in #Reviewer-d489-Q1-A1, our framework can perform lossless information transmission from the regional perception of the environment to the downstream planning task. This endows our UAD with efficient utilization of training data to learn task-relevant knowledge, hence not requiring even backbone pretraining to provide initialization and prior information. Another possible reason lies in the balance of multi-task learning. The numerous optimization losses from preceding subtasks in UniAD would distract the optimization of the oriented planning task in the E2E model. This forces UniAD to load the pretrained weights of BEVFormer to initialize the upstream perception task, which decreases the training difficulty by reducing part optimization items. In contrast, our framework is simple with only five losses for optimization, of which three ones are explicit planning-relevant. The design thus guarantees that the model allocates more attention to the core planning goal without requiring pretrained weights, which is friendly for real-world deployment.
>
> We will include the discussion and analysis in the revision. Thanks.
>
> ***
> ***Q4: It is suggested to provide discussion of limitations and broader impact in the revision.***
>
> **A4**: Thanks for the helpful comment. As discussed in Sec. 4.5 of the manuscript, #Reviewer-d489-Q4-A4, and #Reviewer-3W73-Q4-A4, the coarse perception in our method may occasionally lead to inaccurate planning. However, the flexibility of our simple framework allows the easy integration of customized perception modules (e.g., 3D detection/mapping heads) and the implementation of post-processing pipelines. This adaptability addresses practical considerations in current autonomous driving applications.
>
> Nevertheless, in this way, the need for costly 3D annotations and modular designs persists, posing challenges to the development of efficient end-to-end autonomous driving systems. We believe that in the future, redundant perception and prediction sub-tasks will be optimized or fused in an efficient manner for practical products. We hope our efforts contribute to accelerating this progress.
>
> We will include the discussion in the revision. Again, thanks!

---

> > ### Comment · Reviewer_YXrs · 2024-08-13
> > **Post Rebuttal Comments**
> >
> > Thanks the authors for their detailed feedback. All my concerns have been addressed. I very like this paper due to its novelty, great performance and simplicity. I believe this is a potentially right way for end-to-end autonomous driving.
> >
> > I also see other reviewers' comments and the authors' rebuttals, and think that the authors have done a good job to explain their work better.
> >
> > Thus, I keep my original score (Very Strong Accept).

---

### Official Review · Reviewer_3W73 · 2024-07-12

**Soundness:** 4
**Presentation:** 3
**Contribution:** 3
**Rating:** 8
**Confidence:** 5

**Summary:**

This paper addresses the limitations of current end-to-end autonomous driving models that still rely on modular architectures with manually annotated 3D data. The authors propose an unsupervised pretext task that eliminates the need for manual 3D annotations by predicting angular-wise spatial objectness and temporal dynamics. This is achieved through an Angular Perception Pretext that models the driving scene without the need for manual annotation. A self-supervised training approach is introduced to enhance the robustness of planning in steering scenarios. This strategy learns the consistency of predicted trajectories under different augmented views. UAD demonstrates significant improvements in performance over existing methods like UniAD and VAD in both open-loop and closed-loop evaluations. It achieves these improvements with reduced training resources (44.3% of UniAD) and faster inference speed (3.4× faster than UniAD).

**Strengths:**

1. The paper introduces UAD, an unsupervised autonomous driving framework that eliminates the need for costly 3D manual annotations, which is a significant departure from traditional modular approaches.
2. The Angular Perception Pretext is an innovative approach to spatial-temporal understanding without manual labeling, offering a new perspective on autonomous driving perception.
3. The experiments conducted are comprehensive, including both open-loop and closed-loop evaluations, which demonstrate the method's effectiveness across different scenarios. The paper provides a detailed comparison with state-of-the-art methods like UniAD and VAD, showcasing the improvements in performance metrics, which adds to the quality of the research.

**Weaknesses:**

1. UAD treats an entire sector as occupied when only a part of it contains an object. This seems imprecise. This could potentially lead to less accurate spatial understanding of the environment. In autonomous driving, overly coarse representations might result in the vehicle making less accurate decisions, such as unnecessary braking or incorrect path planning. Have the authors tried some open world segmentation models for more accurate spatial information?
2.  The paper draft does not provide explicit evidence or analysis on whether UAD can indeed benefit from training on a larger scale of data. The authors could conduct experiments with varying sizes of datasets to empirically evaluate how performance metrics change as more data becomes available. This could provide insights into the benefits of scaling up.

**Questions:**

1. it is not explicitly stated whether UAD and UniAD use the exact same training and test split within the nuScenes dataset.
2. As UAD only uses basic obstacles for training.  I wonder how will UAD reacts to traffic lights, lane lines, and policeman's gestures?
3. What is the BEV area designed in the experiment?

**Limitations:**

In the current draft, UAD might be limited to basic obstacle detection and does not extend to the interpretation of traffic signals.

---

> ### Author Rebuttal · Authors · 2024-08-06
>
> ## Response to Reviewer 3W73:
> We appreciate your careful review and thoughtful comments. We are encouraged and grateful that the reviewer found our approach to be well-motivated and innovative. Below, we address the concerns that were raised, and remain committed to clarifying further questions that may arise during the discussion period.
>
> ***
> ***Q1: In autonomous driving, overly coarse representations might result in the vehicle making less accurate decisions, such as unnecessary braking or incorrect path planning. Have the authors tried some open-world segmentation models for more accurate spatial information?***
>
> **A1**: Thanks for the insightful comment.
>
> (1) In the context of human designed planning stack, we agree with the reviewer that inaccurate or overly coarse environment representations can lead to suboptimal or even unsafe vehicle behaviors. We argue that in the context of end-to-end driving stack, this assumption may no longer hold, as our methods have demonstrated the effectiveness, following standard evaluation protocol. We believe that the capability of lossless information transmission can compensate for such coarse representations, as discussed in #Reviewer-d489-Q1-A1.
>
> (2) Different from sending sparse object queries to the planning head, as done in previous works like UniAD and VAD, our model passes regional angular queries to the downstream planning module instead of the predicted objectness probability. Angular queries provide a compact and comprehensive representation of the environment, ensuring lossless transmission of environmental information. In contrast, the object queries used in prior works may miss critical environmental details due to the limitation of detection accuracy.
>
> (3) In the ablation experiment presented in Tab.6 of the manuscript, we explored predicting dense pixel-wise segmentation masks. This comparison may align with and can address the reviewer’s concern. As shown in the results, more precise predictions did not yield better outcomes. We believe this is because the planning module receives more comprehensive and understandable environmental information from angular queries rather than segmentation masks.
>
> (4) Following the suggestion, we also experimented to generate 2D segmentations within GroundingDINO's 2D boxes using the open-set segmentor SAM. We retrained our planner with these 2D segmentations, and the comparison is presented in #Rebuttal-PDF-Tab.2/Fig.2. The results indicate that more fine-grained 2D segmentations offer only marginal performance gains compared with using 2D boxes from GroundingDINO, which further supports our hypothesis. For economic and efficiency reasons, using 2D boxes is more favorable for real-world deployment.
>
> We will include the results and analysis in revision. Thanks, again.
>
> ***
> ***Q2: The authors could conduct experiments with varying sizes of datasets to empirically evaluate how performance metrics change as more data becomes available. This could provide insights into the benefits of scaling up.***
>
> **A2**: Thanks for this constructive comment. As suggested, we trained our UAD model with different volumes of training data, and the results are presented in #Rebuttal-PDF-Tab.4. The comparison shows that even with only 50% of the data, our UAD outperforms the baseline models UniAD and VAD. Increasing the amount of data further enhances the model's performance. It is also observed that more challenging steering scenarios, such as turning right and left, offer greater potential for improvement. We believe that the performance in these challenging scenarios could reach new heights with the availability of more data. This will be a focus of our future research.
>
> We will include the analysis and results in the revision. Thanks.
>
> ***
> ***Q3: It is not explicitly stated whether UAD and UniAD use the exact same training and test split within the nuScenes dataset.***
>
> **A3**: Following the standard protocol in previous works (e.g. UniAD, VAD), we perform training on the 700 scenes and evaluate on the 150 scenes of the nuScenes dataset. Notably, with our proposed unsupervised pretext task, the training process requires no human annotations in nuScenes (e.g., 3D bounding boxes). To make it more clear, we will add clarification on this point in revision. Code, models and configs will be released upon publication. Thanks, again.
>
> ***
> ***Q4: As UAD only uses basic obstacles for training. I wonder how will UAD reacts to traffic lights, lane lines, and policeman's gestures?***
>
> **A4**: Thanks for the thoughtful comments. We follow the paradigm of our baseline UniAD, which does not explicitly send traffic light states or police gestures to the model. Our aim is motivating the E2E planner to understand the world from the data itself, i.e., data-driven learning. Surprisingly, even without the explicit input of such information, #Rebuttal-PDF-Fig.1 shows that our model can correctly interpret the red traffic light and brake the vehicle accordingly, as well as adjusting the driving direction when approaching the lane lines.
>
> We also agree with the reviewer that incorporating such information could help the model understand and react to changes in the environment more easily. An intuitive approach is to transform the traffic light state or other information into queries that interact with the ego planning query. We plan to explore this in future research. Thanks again for the insightful feedback.
>
> ***
> ***Q5: What is the BEV area designed in the experiment?***
>
> **A5**: As mentioned in Section 4.1 of the manuscript, we adopt the view transformer from BEVFormer as the BEV encoder, which sets the BEV region to -51.2m to 51.2m in both the x and y directions. The default BEV resolution is 200 x 200, which aligns with our baseline UniAD. Tab.14 in the appendix of the manuscript lists the performances at different resolutions. We will clarify this in the revision. Thanks.

---

> > ### Comment · Reviewer_3W73 · 2024-08-12
> >
> > Thank you for answering the questions. All my concerns have been addressed. I will keep my initial rating.

---

> > > ### Author Response · Authors · 2024-08-13
> > > **Thank you so much for the feedback!**
> > >
> > > Dear Reviewer 3W73,
> > >
> > > Thank you again for your kind review and constructive comments. Your suggestions greatly strengthen our work and improve our manuscript's quality. We will include them in the revision.
> > >
> > > Best regards,
> > >
> > > Authors

---

### Official Review · Reviewer_UZRG · 2024-07-13

**Soundness:** 2
**Presentation:** 3
**Contribution:** 2
**Rating:** 4
**Confidence:** 5

**Summary:**

This paper aims to discard the requirement of 3D manual annotation in end-to-end autonomous driving by the proposed angular perception pretext task. Besides, this paper proposes a direction-aware learning strategy consisting of directional augmentation and directional consistency loss. Finally, the proposed method UAD achieves superior performance in both open-loop and closed-loop evaluation compared with previous vision-based methods with much lower computation and annotation costs.

**Strengths:**

1) This paper aims to discard the requirement of 3D manual annotation in end-to-end autonomous driving, which is important and meaningful for training larger end-to-end autonomous driving models at scale. I totally agree and appreciate this.
2) This paper proposes a direction-aware learning strategy, which will further improve prformance by self-supervised learning.
3) UAD is evaluated in both open-loop and closed-loop evaluation and different metrics (UniAD, VAD, and BEV-Planner).

**Weaknesses:**

1) There is a lack of explanation on how to use ego status. Besides, there should be more experiments about the performance of UAD without ego status.
2) There is a lack of explanation on how many frames are fused and what method is used for temporal fusion (sliding window or streaming).
3) The angular perception pretext task introduces 2D detection information for perception learning on BEV features. According to Table 6, it seems that angular design is very important for UAD. However, BEV-Planner can achieve a not-so-bad result without any 3D manual annotation and 2D detection information. Therefore, verifying the effectiveness of angular design on BEV-Planner will be more convincing.
4) For the proposed angular design, I do not think is very novel. Because the effectiveness of the 2D detection auxiliary head has been verified in BEVFormerV2 [1] and StreamPETR [2], the UAD just converts 2D object detection to BEV segmentation.
5) For the proposed direction-aware learning strategy, although it is useful, it is a method of data augmentation in EBV space, which I do not think is very novel.

**Questions:**

1) I have serious concerns about the use of ego status and temporal fusion. If UAD uses GT ego status when fusing many frames, because there is no cumulative error in the ego trajectory, it will lead to falsely high performance. A better solution is to use predicted ego status instead of GT ego status (such as SparseDrive [3]).
2) Why does the use of a 3D detection head lead to a decrease in performance? What will be the result if UAD uses the online mapping head?
3) What is the performance if UAD uses a 2D detection auxiliary head instead of an angular design? For example, angular queries can be obtained by 2D detection auxiliary head and depth prediction (such as Far3D [4]).


[1] BEVFormer v2: Adapting Modern Image Backbones to Bird's-Eye-View Recognition via Perspective Supervision
[2] Exploring Object-Centric Temporal Modeling for Efficient Multi-View 3D Object Detection
[3] SparseDrive: End-to-End Autonomous Driving via Sparse Scene Representation
[4] Far3D: Expanding the Horizon for Surround-view 3D Object Detection

**Limitations:**

See the weakness and questions.

---

> ### Author Rebuttal · Authors · 2024-08-06
>
> ## Response to Reviewer UZRG:
> We thank the reviewer for providing helpful comments on our work. We provide our responses below to address reviewer’s concerns, and remain committed to clarifying further questions that may arise during discussion period.
> ***
> ***Q1: Lacking explanation on how to use ego status, and experiments without ego status.***
>
> **A1**: As mentioned in Sec.4.2 of the manuscript, we follow previous works (VAD and BEVPlanner) to embed ego status into planning module. In specific, we concatenate the ego states with ego query along channel dimension. Besides, we have claimed that **only** the rhombus mark in Tab.1 of manuscript denotes the version using ego status. The other experiments are all conducted without ego status for fair comparisons. We will clarify this in revision. Thanks.
> ***
> ***Q2: Lacking explanation on how many frames are fused and temporal fusion.***
>
> **A2**: It should be noticed that we have mentioned using the view transformer from BEVFormer to encode BEV features in Sec.4.1 of manuscript, which performs temporal fusion with BEV feature of the last frame through deformable attentions. We will clarify this in revision. Thanks.
> ***
> ***Q3: Applying angular design on BEVPlanner is more convincing.***
>
> **A3**: (1) It's noted that BEVPlanner claims good performance with only ego status due to dominant "go straight" scenarios in nuScenes. However, in challenging steering scenarios, using solely ego status is insufficient, as also proved in BEVPlanner. In contrast, our UAD achieves superior performance across all scenarios, even without ego status.
>
> (2) Our **ego status version** can be seen as BEVPlanner with our module, since there is no big difference in the planning head. We wanted to follow reviewer's advice to apply our design on BEVPlanner. However, (a) their code was not released when we submitted our paper; and (b) although some code recently released, there was no README in repository until now, making it hard to reproduce official results. We will attempt this when complete code and training guidelines are released. Thanks.
> ***
> ***Q4: Angular design is not very novel as a 2D detection auxiliary head.***
>
> **A4**: The authors would like to re-state the core idea and contributions of the paper. Our work mainly aims to clarify (1) the unnecessity of complex auxiliary tasks for E2E models and (2) E2E models can achieve impressive results even without 3D annotations. Designing simple unsupervised pretexts embodies the implementation of our spirit, where specific module structure in the paper is not the only choice. Hence, simplicity and effectiveness are much more crucial to our work.
>
> Moreover, our work fundamentally differs from papers that utilize 2D detection priors for 3D detection tasks. In our case, 2D detection is applied offline and only used to generate labels for angular perception, which is still conducted in 3D BEV space. This is entirely different from adding an auxiliary 2D detection head. Besides, the predicted objectness is not passed to downstream tasks. We will clarify this in revision. Thanks.
> ***
> ***Q5: Direction-aware learning is not very novel as a data augmentation.***
>
> **A5**: It should be noticed that we greatly alleviated the previous unsolved data bias issue in E2E driving problems by this design. As noted in Fig.2 of BEVPlanner, ego trajectories from E2E training set have highly skewed distributions with most simple go-straight scenarios. Yet, there are no **efficient** solutions in recent E2E works for such severe data bias from the perspective of data augmentation. We then design direction-aware learning to attempt at solving the bias in an efficient data augmentation manner. Despite its simplicity, we believe such finding and strategy can greatly benefit the E2E driving research community. Thanks.
> ***
> ***Q6: Using predicted ego status is better than GT ego status for causing false high performance.***
>
> **A6**: Thanks for this constructive comment. (1) For the non-interactive open-loop evaluation in nuScenes, we follow the way using ego status of BEVPlanner to ensure a fair comparison. (2) The ego status for the closed-loop evaluation in CARLA indeed uses the predicted values provided by the IMU sensor of the ego car, which satisfies the requirement of reviewer. Please refer to #Reviewer-XcsQ-Q1-A1 for updated closed-loop results with ego status. Thanks.
> ***
> ***Q7: Why 3D detection head lead to performance decrease? Applying online mapping head?***
>
> **A7**: (1) The optimization of an additional 3D detection task might distract planning-oriented learning, which is also claimed by UniAD. For instance, the planning performance when training five task heads jointly is much lower compared with planning-oriented training, which transfers features from preceding subtasks to planning (#0 vs #12 in Tab.2 of UniAD). For more discussion, please refer to #Reviewer-d489-Q1-A1. (2) As suggested, we arranged an additional map head to our model as shown in #Rebuttal-PDF-Tab.3, which doesn't improve planning performance.
>
> Notably, in our manuscript, we highlight the "convenience" of integrating traditional perception tasks with our model, not to enhance planning quality but to demonstrate flexibility. Our work aims to, and successfully proves, that these typical perception tasks are not necessary for E2E planning. Thanks.
> ***
> ***Q8: Using a 2D detection auxiliary head instead of angular design?***
>
> **A8**: As discussed in #Reviewer-d489-Q1-A1, lossless information transmission should be the core principle of E2E models. Passing 2D object queries to planning module is no different from previous works using 3D object queries, which miss important environmental information outside the scope of human-defined object categories. Hence our model performs angular perception primarily to summarize comprehensive environmental information from BEV feature, rather than using its output as in previous works. We will clarify this in revision. Thanks.

---

> > ### Comment · Reviewer_UZRG · 2024-08-13
> >
> > Thanks for your response.  However, after reading the author's rebuttal, some issues still need to be addressed, especially the experiments.
> > 1. Experiments without ego status are required. The BEV-Planner contains relevant experiments without ego status.
> > 2. How many frames are fused? Is streaming time fusion used to fuse many frames?
> > 3. I don't think angular perception is very different from 2D detection and depth estimation. Angular perception is just projecting the 2D detection box into BEV, which only involves the additional operation of extrinsic parameter transformation.

---

> > > ### Author Response · Authors · 2024-08-13
> > > **Response to Reviewer UZRG**
> > >
> > > Thank you for taking the time to offer suggestions amidst your busy schedule, yet we still need to point out several misunderstandings by the reviewer about our rebuttal. The detailed responses are provided below, hoping to address your concerns adequately:
> > >
> > > ***
> > > ***Discussion-Q1: Experiments without ego status are required. The BEV-Planner contains relevant experiments without ego status.***
> > >
> > > **Discussion-A1**: As clarified in #Rebuttal-Q1-A1, **only the rhombus mark** in Tab.1 of the manuscript denotes the version using ego status. The other experiments are **all** conducted **without ego status** for fair comparisons. Our planner achieves the best open-loop performance in nuScenes for both w/. and w/o. ego-status settings, as shown in Tab.1 of the manuscript. We will clarify this in the revision. Thanks.
> > >
> > >
> > > ***
> > > ***Discussion-Q2: How many frames are fused? Is streaming time fusion used to fuse many frames?***
> > >
> > > **Discussion-A2**: As clarified in #Rebuttal-Q2-A2, we use the view transformer from BEVFormer[1] to encode BEV features, which performs temporal fusion with the BEV feature of the last frame through deformable attentions. In other words, there are total 2 frames for temporal fusion, i.e. t-1 and t. For more details, please refer to [1]. We will clarify this in the revision. Thanks.
> > >
> > > [1] Li Z, et al. Bevformer: Learning bird’s-eye-view representation from multi-camera images via spatiotemporal transformers. In ECCV 2022.
> > >
> > > ***
> > > ***Discussion-Q3: I don't think angular perception is very different from 2D detection and depth estimation. Angular perception is just projecting the 2D detection box into BEV, which only involves the additional operation of extrinsic parameter transformation.***
> > >
> > > **Discussion-A3**: In #Rebuttal-Q4-A4, we have detailed the fundamental differences between our angular perception and 2d detection/depth estimation, as well as the core idea and contributions of our work. It should be noted that the projection of 2D boxes is only used to generate the pseudo labels for training our pretext task, instead of 'detecting' the object areas for downstream tasks. **The processes of projection and objectness prediction are not performed during inference, since the angular queries have provided compact and comprehensive environmental knowledge**. We will clarify this in the revision. Thanks.
> > >
> > > ***
> > > Again, we thank the reviewer for the time and efforts, and remain committed to clarifying further questions.

---

> > > > ### Comment · Reviewer_UZRG · 2024-08-14
> > > >
> > > > Thanks for your reply.
> > > >
> > > > For Q1, I think there should be the results of experiments without ego status under the BEV-Planner metrics for a clearer comparison. Besides, UAD uses the structure of BEVFormer, so whether ego status is introduced into the BEV feature like BEVFormer?
> > > >
> > > > For Q2, does this mean that the UAD does not use streaming temporal fusion?
> > > >
> > > > For Q3, 2D detection task is not necessarily applied for downstream tasks. In fact, StreamPETR utilizes the 2D auxiliary head to improve the performance. UAD projects the 2D boxes into the BEV and applies a mask prediction task. It seems to me that only the form of the prediction has changed. And I'm not clear about the difference in performance for angular perception and 2D auxiliary head, since authors do not provide the experiments in Q8 of the previous rebuttal.

---

> > > > > ### Author Response · Authors · 2024-08-14
> > > > > **Second Response to Reviewer UZRG**
> > > > >
> > > > > Thanks again for your feedback. We provide the responses below, hoping to address your concerns adequately:
> > > > >
> > > > > ***
> > > > > ***Discussion-re-Q1: Whether ego status is introduced into the BEV feature like BEVFormer? Experiments without ego status under the BEV-Planner metrics.***
> > > > >
> > > > > **Discussion-re-A1**: (1) The default version of our method in the manuscript follows BEVFormer, which utilizes "ego status" to align BEV features across different frames for temporal fusion.
> > > > >
> > > > > (2) Based on the reviewer’s suggestion, we removed **all "ego status" usage** from our framework. In this scenario, the temporal fusion module within the view transformer degrades to single-frame variant (#2 in the table below) and simple-concatenate variant (#4 in the table below, where we simply concatenate the BEV features of two adjacent frames without alignment). Despite these modifications, our UAD still achieves superior performance **under all settings and metrics** (i.e., #1 vs. #2 and #3 vs. #4).
> > > > >
> > > > > We will include the results and analysis in the revision. Again, thank you for your valuable feedback!
> > > > >
> > > > > **Table 1:** Ablation on the ego status and temporal fusion. Col. and Int. denote collision rate and intersection rate with road boundary, respectively. **All metrics are aligned with BEV-Planner**.
> > > > > |# | Method | Ego Status ***in BEV*** | Ego Status ***in Planner*** | Temporal Fusion |  | L2-1s | L2-2s | L2-3s | Avg. L2 || Col.-1s | Col.-2s | Col.-3s | Avg. Col. || Int.-1s | Int.-2s | Int.-3s | Int. Col. |
> > > > > | :-: | :-: | :-: | :-: | :-: | :-: | :-: | :-: | :-: | :-: | :-: | :-: | :-: | :-: | :-: | :-: | :-: | :-: | :-: | :-: |
> > > > > |1| BEV-Planner | No | No | No | |0.27|0.54|0.90|0.57| |0.04|0.35|1.80|0.73| |0.63|3.38|7.93|3.98|
> > > > > |2| UAD (Ours) | No | No | No | |0.25|0.43|0.76|0.48| |0.02|0.26|1.23|0.50| |0.37|1.53|4.48|2.13|
> > > > > |3| BEV-Planner | No | No | Yes | |0.30|0.52|0.83|0.55| |0.10|0.37|1.30|0.59| |0.78|3.79|8.22|4.26|
> > > > > |4| UAD (Ours) | No | No | Yes | |0.22|0.39|0.67|0.43| |0.01|0.21|0.85|0.36| |0.18|1.16|3.51|1.62|
> > > > > |5| BEV-Planner | Yes | No | Yes | |0.28|0.42|0.68|0.46| |0.04|0.37|1.07|0.49| |0.70|3.77|8.15|4.21|
> > > > > |6| UAD (Ours) | Yes | No | Yes | |0.21|0.35|0.60|0.39| |0.01|0.17|0.78|0.32| |0.16|1.12|3.46|1.58|
> > > > > |7| BEV-Planner | Yes | Yes | Yes | |0.16|0.32|0.57|0.35| |0.00|0.29|0.73|0.34| |0.35|2.62|6.51|3.16|
> > > > > |8| UAD (Ours) | Yes | Yes | Yes | |0.13|0.28|0.48|0.30| |0.00|0.12|0.55|0.22| |0.10|0.80|2.48|1.13|
> > > > >
> > > > > ***
> > > > > ***Discussion-re-Q2: Does this mean that the UAD does not use streaming temporal fusion?***
> > > > >
> > > > > **Discussion-re-A2**: The temporal fusion in the view transformer of BEVFormer is categorized as streaming temporal fusion, where the BEV features from the past frame are fused into the next frame. We adhere to this pipeline in our UAD method. We will clarify this point in the revision. Thank you for your insightful feedback.
> > > > >
> > > > > ***
> > > > > ***Discussion-re-Q3: (1) 2D detection task is not necessarily applied for downstream tasks. In fact, StreamPETR utilizes the 2D auxiliary head to improve the performance. UAD projects the 2D boxes into the BEV and applies a mask prediction task. It seems to me that only the form of the prediction has changed. (2) And I'm not clear about the difference in performance for angular perception and 2D auxiliary head, since authors do not provide the experiments in Q8 of the previous rebuttal.***
> > > > >
> > > > > **Discussion-re-A3**: Thanks for the comment.
> > > > >
> > > > > (1) We agree that the output of auxiliary tasks is not necessary for downstream ones. However, the key difference between the mentioned 2D auxiliary head and our angular perception lies in our primary objective: we designed the model to extract compact environmental knowledge using angular queries instead of just improving performances with auxiliary tasks. In other words, our spatial module provides a new approach to summarize environmental information in the E2E framework.
> > > > >
> > > > > (2) The angular queries in our method effectively summarize the environmental information, ensuring lossless information transmission. When using 2D auxiliary heads and the sparse queries derived from these heads, we do not observe significant differences compared to existing 3D detection heads. This is why we argue that the results would likely be worse than those achieved with a 3D auxiliary detection head (please refer to Table 8 of the manuscript). Moreover, our temporal representation learning is specifically tailored to the angular design, enabling the prediction of future dynamics within each sector region: a task that cannot be performed effectively with 2D object queries.
> > > > >
> > > > > Again, we emphasize that our core philosophy is to eliminate the need for costly modularized designs and manual annotations.
> > > > >
> > > > > ***
> > > > > We sincerely thank the reviewer for the time and efforts, and remain committed to clarifying further questions.

---

> ### Author Response · Authors · 2024-08-11
> **Discussion Invitation**
>
> Dear Reviewer UZRG,
>
> We thank you for the precious review time and valuable comments. We have provided corresponding responses and results, which we believe have covered your concerns. We hope to further discuss with you whether or not your concerns have been addressed. Please let us know if you still have any unclear parts of our work.
>
> Sincerely,
>
> Authors.

---

> ### Comment · Reviewer_UZRG · 2024-08-14
>
> Thanks for your response. I only raise my score to 4. We are still confused about the design of angular perception, and this paper needs a lot of revisions to clearly describe these details, especially the use of ego status and temporal information.

---

### Official Review · Reviewer_XcsQ · 2024-07-14

**Soundness:** 3
**Presentation:** 4
**Contribution:** 2
**Rating:** 3
**Confidence:** 5

**Summary:**

The article proposes an end-to-end (E2EAD) autonomous driving method called UAD (Unsupervised Autonomous Driving), which achieves autonomous driving on a visual basis without the need for expensive modular design and 3D manual annotation. UAD aims to overcome the limitations of existing E2EAD models that mimic traditional driving stack module architectures. These models typically require carefully designed supervised perception and prediction subtasks to provide environmental information for planning, which require a large amount of high-quality 3D annotation data and consume significant computational resources during training and inference processes.

**Strengths:**

1. The method is novel and a good direction for exploring the end-to-end model's dependence on 3D manual annotation.
2. The paper has rich ablation experiments to demonstrate the effectiveness of the method.
3. The paper has advantages in both speed and accuracy compared to previous articles

**Weaknesses:**

1. The paper lacks sufficient comparison with other methods(such as Interfuser[1],DriveAdapter[2],DriveMLM[3],VADv2[4]), which acheve better closed-loop performance on carla town 05 long benchmark.
2. In Table 6, the angular design brings too much gain, especially in terms of collision rate (from 1.37% to 0.19% ). It's strange. The angular design is about how to encode the sensor data and should not have so much impact on collision rate.
3. Angular design is widely used in BEV-related works, like PolarFormer[5]. The paper lacks citation about these works.And it's not proper to regard it as the main contribution of the work.
4. The paper lacks a part to introduce the use of 2D tasks as auxiliary tasks.

[1] Hao Shao, Letian Wang, Ruobing Chen, Hongsheng Li, and Yu Liu. Safety-enhanced autonomous driving using interpretable sensor fusion transformer. In Conference on Robot Learning, pages 726–737. PMLR, 2023
[2] Xiaosong Jia, Yulu Gao, Li Chen, Junchi Yan, Patrick Langechuan Liu, and Hongyang Li. Driveadapter: Breaking the coupling barrier of perception and planning in end-to-end autonomous driving. 2023
[3] Wenhai Wang, Jiangwei Xie, ChuanYang Hu, Haoming Zou, Jianan Fan, Wenwen Tong, Yang Wen, Silei Wu, Hanming Deng, Zhiqi Li, et al. Drivemlm: Aligning multi-modal large language models with behavioral planning states for autonomous driving. arXiv preprint arXiv:2312.09245, 2023
[4] Shaoyu Chen, Bo Jiang, Hao Gao, Bencheng Liao, Qing Xu, Qian Zhang, Chang Huang, Wenyu Liu, and Xinggang Wang. Vadv2: End-to-end vectorized autonomous driving via probabilistic planning. arXiv preprint arXiv:2402.13243, 2024
[5] Yanqin Jiang, Li Zhang, Zhenwei Miao, Xiatian Zhu, Jin Gao, Weiming Hu, Yu-Gang Jiang. PolarFormer: Multi-camera 3D Object Detection with Polar Transformer. AAAI 2023

**Questions:**

Why use 2D boxes instead of 2D segmentation of objects? Is it more reasonable to use 2D segmentation labels to link the points in BEV space with the points in image space?

**Limitations:**

1. Incompele comparison on CARLA benchmark.
2. Lack citation about angular design.
3. Angular design is not novel.
4. Some of the experiment results are not that convincing.

---

> ### Author Rebuttal · Authors · 2024-08-06
>
> ## Response to Reviewer XcsQ:
> We thank the reviewer for providing thoughtful comments on our work. We provide our responses below to address reviewer’s concerns, and remain committed to clarifying further questions that may arise during discussion period.
>
> ***
> ***Q1: The paper lacks sufficient comparison with other methods such as ...VADv2.***
>
> **A1**: Thanks for this helpful comment. Recent works, including VAD and some papers mentioned by the reviewer, don't provide their code and models for CARLA closed-loop evaluation. Therefore, in the comparison of the manuscript, we **don't use ego-status** (in the planning module) as we want to guarantee fair comparison with our closed-loop baseline TransFuser, which releases the code and is a standard closed-loop framework.
>
> We then construct fair comparisons with mentioned works by providing a **using ego-status version** in #Rebuttal-PDF-Tab.1, which again proves its effectiveness. Particularly, our model outperforms all mentioned methods except the driving scores of VADv2. Notably, no 3D manual annotation is needed with such a SOTA performance. Since VADv2 estimates multi-modal ego trajectories while our work (also our baselines) predicts single-modal one, we believe our UAD can also benefit from the multi-modal design, as it has been proved by motion prediction works. Yet this study is not the topic of our work, we leave it to future study.
>
> We will include the results in revision. Thanks.
>
> ***
> ***Q2: The angular design should not have so much impact on collision rate.***
>
> **A2**: Thanks for the comment, yet we respectfully disagree. As discussed in #Reviewer-d489-Q1-A1, the E2E frameworks provide potential of lossless information transmission from sensors to planning head, compared with previous modularized design. However, planners in most prior works perceive the environment from sparse object queries with limited and manually defined categories (e.g., vehicles). Any information not covered by these categories or related perception subtasks is inevitably lost. Moreover, the errors from upstream sub-tasks like object detection would directly hinder the performance of downstream planning. In contrast, our angular queries contain compact and lossless information about the driving scene, motivating the E2E model to make more intelligent planning decisions, especially after training with large-scale datasets.
>
> In addition to the analyses above, we demonstrated our claim with experiments, as Tab.4 and Tab.6 in the manuscript have clearly evidenced that both our spatial and temporal perception designs can decrease the collision rate. We will include the analysis in the revision. Thanks.
>
> ***
> ***Q3: Angular design is widely used like PolarFormer and not proper as main contribution.***
>
> **A3**: Thanks for the comment. To the best of our understanding, our angular design non-trivially differs with previous works like PolarFormer, which builds grid-wise BEV feature by **projecting** polar coordinates instead of Cartesian coordinates to 2D images. Yet, the essence of our angular design, is not the process of building BEV feature. Instead, we aim to explore effective strategies to **losslessly transfer the information** of the BEV feature for planning, i.e., how to **summarize** the driving scene from BEV grids. We believe our angular design is both compact and efficient, which is compatible with different BEV construction paradigms such as LSS, PolarFormer or BEVFormer. How to build better BEV features, and how to pass BEV features in a compact and lossless manner to downstream tasks, are orthogonal research directions in the larger E2E driving context. We believe our core novelty is the spirit of relaxing costly modularizations and removing manual annotation. Again, we thank the reviewer for this comment and will add clarification in revision to make it more clear.
>
> ***
> ***Q4: The paper lacks a part to introduce 2D auxiliary tasks.***
>
> **A4**: Thanks for the helpful comment. With the spatial relationship between the multi-view images and BEV space, a few works exploit 2D tasks to provide auxiliary clues for accurate BEV perception. For instance, MV2D exploits 2D detectors to generate object queries conditioned on rich image semantics, which help to recall objects in camera views and localize 3D objects. Far3D similarly utilizes a 2D detector and a depth predictor to generate reliable 2D box proposals and their corresponding depths, which are then concatenated and projected into 3D space as object anchors to predict the locations.
>
> Notably, the intermediate results from the auxiliary tasks, e.g., 2D boxes, are used for the downstream tasks in the aforementioned works. But differently, the mask from our introduced angular perception is not used, since we tend to losslessly transfer the environment information from the pretext to the planning module.
>
> We will include the discussion in revision. Thanks.
>
> ***
> ***Q5: Why use 2D boxes instead of 2D segmentation of objects?***
>
> **A5**: Thanks for the constructive comment. (1) The design concept of our work is to provide crucial environmental knowledge while avoiding heavy annotation and computation costs. Compared with easily achievable 2D boxes, the 2D segmentations undoubtedly bring more overload. (2) Since the BEV grids are sparsely distributed, there are no clear differences after projecting the grids in a sector to the 2D boxes or segmentation masks and then summarizing their information as pseudo labels.
>
> As suggested, we also try to generate 2D segmentations within the 2D boxes by the open-set segmentor SAM. Then we retrain our planner with the 2D segmentations, and the results are listed in #Rebuttal-PDF-Tab.2/Fig.2. It shows that more fine-grained 2D segmentations only bring small performance gains compared with the one using 2D boxes. For economic and efficiency reasons, applying 2D boxes is more friendly for real-world deployment.
>
> We will include the results and discussion in revision. Thanks.

---

> ### Author Response · Authors · 2024-08-11
> **Discussion Invitation**
>
> Dear Reviewer XcsQ,
>
> We thank you for the precious review time and valuable comments. We have provided corresponding responses and results, which we believe have covered your concerns. We hope to further discuss with you whether or not your concerns have been addressed. Please let us know if you still have any unclear parts of our work.
>
> Sincerely,
>
> Authors.

---

### Official Review · Reviewer_d489 · 2024-07-17

**Soundness:** 3
**Presentation:** 3
**Contribution:** 3
**Rating:** 6
**Confidence:** 5

**Summary:**

The paper proposes a new e2e driving model named UAD. In this paper, the authors propose an unsupervised method for effective training and inference of the e2e model. The paper mainly has two contributions: 1. It designs an angular-wise perception module. In this module, the authors directly project 2D GT labels onto the BEV and define a new BEV map label for perception training. This module design can efficiently reduce complexity and preserve effectiveness. 2. The authors propose a direction-aware method to augment the trajectory training and use consistency loss for further supervision.

The final results show the effectiveness and soundness of the proposed method.

**Strengths:**

1. The idea of the angular-wise perception module is interesting. It can utilize a huge amount of 2D annotated autonomous driving datasets to train the e2e model, which removes the restriction of the limited number of 3D annotations.
2. The proposed direction-aware method for trajectory prediction is also meaningful since it can add additional consistency loss for better supervision.
3. The results are promising and the efficiency improvement is impressive.

**Weaknesses:**

1. The design of the angular-wise perception module is a little bit counter-intuitive to me. From my perspective, it works because (1) It can greatly enlarge the size of the training data. (2) It makes the perception task simpler, thus the model can do it better (knowing an object in a direction is much simpler than detecting the BBox). (3) The efficiency improves because of the light design of the perception task. I think it can be simply treated as a low-resolution object detection task without depth. Except for (1), I do not understand why it can improve the final results.

2. The design of the direction-aware planning training strategy is effective but simple. I cannot see too much insight here. Could the authors provide more insight into this? Or is this just an engineering trick?

3. For experiments, do you use exactly the same training data for both the open-loop and close-loop experiments? If yes, could you provide more analysis about why the results are surprisingly good even if you use a low-resolution detection module? If not, could you provide details about the pertaining data info? Can the impressive results come from leaked data? How to prevent testing data leakage?

4. In real-world applications, when the vehicle plans its path, it needs to "see" a lot of things, including objects and some other elements, like traffic lights, traffic signals, or some special marks on the road. How can you handle these elements based on your model? For example, can your model understand traffic signals and see red traffic lights? If not, how to extend your model to real-world scenarios and applications?

**Questions:**

Please see the weakness part for questions and suggestions.

**Limitations:**

My main concern with this paper mainly comes from the insight of the angular-wise perception module. I still cannot understand why it works except for the huge amount of additional training data. Provide more details for this.

Besides, how to deploy and extend the model to real-world cases, that requires depth information (e.g., ) or semantic information (e.g., traffic light)? The authors should provide more discussion about this to make the work more promising.

---

> ### Author Rebuttal · Authors · 2024-08-06
>
> ## Response to Reviewer d489:
> We thank the reviewer for providing valuable and thoughtful comments on our work. We provide our responses below to address the reviewer's concerns, and remain committed to clarifying further questions that may arise during the discussion period.
>
> ***
> ***Q1: The working reasons of the angular-wise perception module to improve final results.***
>
> **A1**: Thanks for this insightful comment. Besides all the advantages mentioned by the reviewer, there are two main reasons that we believe can explain the effectiveness of our design:
>
> (1) **Lossless Information Transmission.** The essential difference between the E2E paradigm and the previous modularized design, is that E2E approaches enable the possibility to transmit information without loss, from the input sensors to the planning module in the model architecture. However, planners in UniAD or most other works perceive the environment from sparse object queries with limited and manually defined categories (e.g., vehicles, pedestrians, maps). Any information not covered by these categories or related perception subtasks is inevitably lost. Moreover, the errors from upstream sub-tasks like object detection would directly hinder the performance of downstream planning. On the contrary, the angular queries in our method contain compact and lossless information about the driving environment, which benefits the E2E model in making more intelligent planning decisions, especially after training with large-scale datasets.
>
> (2) **Less is More.** It's a common wisdom that the training of deep learning models is a process of balancing losses of different tasks towards Nash Equilibrium. Complicated designs from previous works make their training process unnecessarily complex and fragile. For example, UniAD needs careful pretraining on early modules. In addition, more sub-tasks can easily distract the E2E model from the planning objective. The results of previous work UniAD evidence this claim. Particularly, the introduction of "Track & Map & Motion & Occ" tasks in a multi-task learning (MTL) manner even degrades the planning performance (#0 v.s. #10 in Tab.2 of the UniAD official paper). Our simpler spatial perception design thus guarantees the model to allocate more attention to the core planning goal, which we believe is a representative practice of Occam's Razor.
>
> We will include the discussion in the revision. Again, thanks.
>
> ***
> ***Q2: The direction-aware training strategy is effective but simple, it seems not having too much insight. Could the authors provide more insight into this?***
>
> **A2**: The reason that the direction-aware training strategy being extremely effective is that we greatly alleviated the previous unsolved data bias issue in E2E driving problems. As noted in BEVPlanner[1] (see Fig.2 of BEVPlanner), ego trajectories from E2E training set have highly skewed distributions: in most scenarios the self-driving vehicles simply go straight. Yet, there are no **efficient** solutions in recent E2E works to mitigate such severe data bias problem from the perspective of data augmentation. We then design the direction-aware training strategy, which attempts at solving the data bias problem in an efficient data augmentation manner. Despite its simplicity, we believe such finding and the corresponding strategy can greatly benefit the E2E driving research community.
>
> [1] Li Z, et al. Is ego status all you need for open-loop end-to-end autonomous driving. In CVPR 2024.
>
> ***
> ***Q3: (1) If the training data for both open-/close-loop experiments are the same? (2) Why the results are surprisingly good even if using a low-resolution detection module? (3) If the impressive results come from leaked data?***
>
> **A3**: (1) The evaluation of open-/closed-loop performance is performed on two **different** benchmarks. In particular, nuScenes only supports open-loop evaluation, and CARLA supports closed-loop testing. Therefore, following the standard protocol in previous works (e.g. UniAD, VAD), we train the model on corresponding data respectively, i.e., train with nuScenes for open-loop evaluation and with CARLA for closed-loop one.
>
> (2) As discussed in Q1, we believe lossless information transmission and the simpler structure for optimization are essential factors for our good improvement. Besides, both **spatial** angular perception and **temporal** latent world-model are the impulses of impressive results.
>
> (3) We adopt the same training and testing pipelines following our baseline UniAD, which we promise no data leakage. Code and models for both open- and closed-loop experiments will be released.
>
> We thank the reviewer for this comment and will add clarification in revision to make it more clear.
>
> ***
> ***Q4: If the proposed planner could understand traffic lights or signals, and how to extend to real-world scenarios and applications?***
>
> **A4**: Thanks for the thoughtful comments. Recent E2EAD models are mostly designed to implicitly perceive and understand object-irrelevant elements like traffic lights and signals through data-driven learning. We follow this paradigm and illustrate the capability in the #Rebuttal-PDF-Fig.1. The visualization shows that our UAD can correctly understand the red traffic light and then brake the vehicle.
>
> As discussed in Sec. 4.5 of our manuscript, it's easy to apply customized tasks like object detection, lane segmentation, also the mentioned traffic light detection that are required in application products under current auto-driving technical conditions to our framework. The intermediate results can be used for post-processing and refinement of the output trajectories from the E2E planner. However, we believe that someday the redundant perception and prediction sub-tasks will be optimized or fused in an efficient manner for practical products. We hope our efforts can speed up this progress.
>
> We will include the results and discussion in the revision. Thanks.

---

> > ### Comment · Reviewer_d489 · 2024-08-13
> > **Final comments**
> >
> > Thank the authors for the detailed response and information. Some of my concerns are addressed, but I still have problems with the design and working principles of the angular-wise perception module. I have strong concerns about its actual application values in real-world scenes. I will slightly raise the score, and the authors are encouraged to further refine the paper and make it more solid.

---

> > > ### Author Response · Authors · 2024-08-13
> > > **Thank you so much for the feedback!**
> > >
> > > Dear Reviewer d489,
> > >
> > > We are sincerely grateful for your decision to raise the score. Your suggestions have greatly enhanced our paper and inspired our future research direction.
> > >
> > > Regarding the application of our work in the real world, (1) we share the same expectation as the reviewer in extending our work to autonomous vehicles. And We are also actively working on implementing our approach in real-world auto-vehicles.  (2) We provided a transition plan in the paper to integrate our approach into current autonomous driving systems. Specifically, our framework can easily accommodate a typical perception model for post-processing. (3) We believe our paradigm will unlock more potential from data. Given that billions of data points are used in real-world scenarios to ensure the robustness of autonomous driving, it is nearly impossible to precisely annotate all the data. Therefore, it is essential to advance efficient and unsupervised end-to-end research. In addition to addressing real-world application needs, we aim to set a potential direction for unsupervised methods in the future. We are grateful that the reviewer recognizes our contribution.
> > >
> > > Once again, thank you for your positive feedback and valuable insights! Your suggestions regarding the deeper exploration of working mechanisms, practical considerations, and experimental details will greatly strengthen our work and improve the quality of our manuscript. We will incorporate them in the revision. Code and models will also be released to facilitate future research.
> > >
> > > Best regards,
> > >
> > > Authors

---

### Author Rebuttal · Authors · 2024-08-06

**Dear Reviewers,**

We thank all reviewers for their very careful review, valuable comments and suggestions on our manuscript. We have worked our best to address all concerns with analyses and experiments according to the comments from reviewers.

In specific, for **Reviewer d489**'s comments, we analyze the working reasons of our angular perception (**Q1**) and provide more insights about the direction-aware learning strategy (**Q2**). In addition, we clarify the date settings in our experiments following the baselines (**Q3**). Furthermore, we show the implicit perception capability for traffic lights or signals by visualization, and discuss the extension to real-world scenarios and applications (**Q4**).

For **Reviewer XcsQ**'s comments, we update the closed-loop evaluation performance by adding ego status for fair comparisons with the mentioned planners (**Q1**). We analyze the working reasons of our angular design to collision rate (**Q2**) and clarify the difference compared with PolarFormer (**Q3**). We also review the use of 2D auxiliary tasks and explain the difference with our pretext (**Q4**). In addition, we explore the influence of 2D ROI form by replacing boxes with segmentations (**Q5**).

For **Reviewer UZRG**'s comments, we clarify the use of ego status (**Q1**,**Q6**), historical frames and temporal fusion (**Q2**). We again explain the working reason of BEVPlanner in nuScenes open-loop evaluation (**Q3**). We also discuss the difference between our angular design and 2D detection auxiliary head (**Q4**,**Q8**), the significance of direction-aware learning (**Q5**). Moreover, we analyze the influence of introducing 3D detection head or online mapping head (**Q7**).

For **Reviewer 3W73**'s comments, we explain the design concept of angular perception for regional environmental knowledge, and try using segmentations as ROIs for precise spatial information (**Q1**). We explore the influence of different volumes of training data (**Q2**). We also clarify the date settings in our experiments following the baselines (**Q3**). Moreover, we show the implicit perception capability for traffic lights or signals by visualization (**Q4**), and detail the configurations of the BEV area following the baseline UniAD (**Q5**).

For **Reviewer YXrs**'s comments, we try different methods to generate 2D ROIs and analyze the performances (**Q1**). We also explore the influence of different volumes of training data (**Q2**). Besides, we analyze the reason of achieving excellent performance even without backbone pretraining, compared with the baseline UniAD (**Q3**). Finally, we discuss the limitations and broader impact of our work (**Q4**).

**For more details, please check individual responses**. We thank all reviewers for their time and efforts! We hope our responses have persuasively addressed all remaining concerns. Please don’t hesitate to let us know of any additional comments or feedback.

**Note that we include all additional experimental results in the one-page PDF submitted along with this global rebuttal response**.

---

### Comment · Area_Chair_E2Wk · 2024-08-12
**Help Check the Rebuttal, Make Discussions, and Update the Final Recommendations**

Dear Reviewers,

Thanks for serving as a reviewer for the NeurIPS. The rebuttal deadline has just passed.

The author has provided the rebuttal, could you help check the rebuttal and other fellow reviewers' comments, make necessary discussions, and update your final recommendations as soon as possible?

Thank you very much.

Best,

Area Chair

---

### Decision · Program_Chairs · 2024-09-25

**Decision:**

Reject

**Comment:**

This paper was reviewed by five experts in the field. The recommendations are diverse (Very Strong Accept, Strong Accept, Weak Accept, Borderline Reject, Reject). Three Reviewers (d489, XcsQ, and UZRG) raise serious concerns regarding the Angular design and angular perception pretext, which share similar insights with previous BEV-related works with limited technical novelty and innovation. Reviewer d489 still has strong concerns about its actual application values in real-world scenes. Reviewer XcsQ argues that the experimental evaluations are not sufficient and convincing, and more detailed ablations and comparisons are needed to validate the effectiveness. Besides more supportive experiments, Reviewer UZRG also claims that a lot of revisions are needed to clearly describe these details. Reviewers 3W73 and YXrs hold positive opinions regarding the manuscript. However, the rebuttal did not assuage all these concerns raised by the former three reviewers. Considering the reviewers' concerns, we regret that the paper cannot be recommended for acceptance at this time. The authors are encouraged to consider the reviewers' comments when revising the paper for submission elsewhere.